# Snow Avalanche Frequency Estimation (SAFE): 32 years of monitoring remote avalanche depositional zones in high mountains of Afghanistan

Arnaud Caiserman[1*], Roy C. Sidle[1], Deo Raj Gurung[2]

[1] Mountain Societies Research Institute – University of Central Asia, Khorog, 736000, Tajikistan

A. Caiserman ORCID: https://orcid.org/0000-0003-4580-6633 ; corresponding author: arnaud.caiserman@ucentralasia.org

Roy C. Sidle ORCID: https://orcid.org/0000-0002-5004-4154

[2] Aga Khan Agency of Habitat, Dushanbe, 734013, Tajikistan

*Correspondence to*: Arnaud Caiserman (arnaud.caiserman@ucentralasia.org)

**Abstract.** Snow avalanches are the predominant hazards in winter in high elevation mountains. They cause damage to both humans and assets but cannot be accurately predicted. Here we show how remote sensing can accurately inventory large avalanches depositional zones every year in a large basin using a 32-yr snow index derived from Landsat satellite archives. This Snow Avalanche Frequency Estimation (SAFE) built in an open-access Google Engine script maps snow hazard frequency and targets vulnerable areas in remote regions of Afghanistan, one of the most data-limited areas worldwide. SAFE correctly detected of the actual depositional zones of avalanches identified on Google Earth and in the field (Probability of Detection 0.77 and Positive Predictive Value 0.96). A total of 810,000 large depositional zones of avalanches occurred since 1990 within an area of 28,500 km$^2$ with a mean frequency of 0.88 avalanches$^{km^2y-1}$, damaging villages and blocking roads and streams. Snow avalanche frequency did not significantly change with time, but a northeast shift of these hazards was evident. SAFE is the first robust model that can be used worldwide and is especially capable of filling data voids on snow avalanche impacts in inaccessible regions.

## 1. Introduction

Snow avalanches are among the fastest, up to 61 ms$^{-1}$, and therefore most dangerous natural hazards in mountain areas (Louge et al., 2012). Casualties associated with avalanches are numerous; in 2021 alone, 37 fatalities occurred in the US (Colorado Avalanche Information Center, 2021) and 127 in Europe (European Avalanches Warning Services, 2021), but avalanche monitoring is not consistent across the globe. Most remote mountain regions and communities are not systematically monitored for avalanche occurrence. Avalanche surveys amongst remote villages are sparse because regions are uninhabited; however, avalanches can block connecting roads every year since avalanche volumes range from hundreds to several tens of thousand cubic meters (Gubler, 1987). Where weather stations exist, avalanches can be predicted based on snow depth and other weather parameters (Greene et al., 2016). However, the global weather monitoring of mountainous areas is scattered and very sparse in developing nations.

To support these science and government priorities in remote mountain regions, it is necessary to introduce a user-friendly, open-access method that maps snow avalanches on an annual basis across wide areas where internet connection and monitoring systems are not always available. As an example, half of the land surface of Afghanistan is above 2000 m a.s.l. and 80% is mountainous (Asad Sarwar, 2002). Among Central Asian nations, Afghanistan's population is most at risk of avalanche hazards; 22,477 inhabitants at risk compared to 5183 in Tajikistan (Chabot and Kaba, 2016). Particularly, northeast Afghanistan (Badakhshan) is one of the most vulnerable regions, especially from December through March (Mohanty et al., 2019). Several international initiatives have been implemented in Afghanistan to forecast avalanches or assess their risks on local communities. According to USAID, 30,600 buildings are at risk of avalanches in Badakhshan based on daily snow depth measurements (USAID, 2021). The Aga Khan Agency for Habitat (AKAH) collects snow depth data and uses models such as Alpine3D and SNOWPACK to forecast avalanche prone regions in Tajikistan, Afghanistan, and Pakistan (Bair et al., 2020). Other products have been developed, such as avalanche susceptibility and exposure maps (Kravtsova, 1990; Soteres et al., 2020; World Bank, 2017). Another approach is to combine topographic maps and snow data via the RAMMS:AVA models (GFDRR, 2018), but these are not open access. Finally, it is possible to count the number of avalanches in each district as done by the United Nations in their map *Districts Affected by Avalanches* (OCHA-United Nations, 2012), but this is time consuming and may miss some events across large areas.

Detecting the avalanches is a challenge and requires temporal as well as spatial data, especially for large areas. Remote sensing technology, both air and spaceborne, can cover large areas at different times of the year. Indeed, the frequent collection of satellite images over the same area enables the detection of changes in snow cover as well as other hazards, such as floods and landslides. Until recently, the use of remote sensing in avalanche detection was sparse due to low resolution, and the automation of such processes was even more difficult because of the lack of relevant algorithms that can compute big data (Eckerstorfer et al., 2016). Other remote sensing approaches for avalanche detection have used radar, Lidar, and optical data. Radar satellites, such as Sentinel-1A and B, are now commonly used for detecting mass movements by assessing backscatter signal changes between two time periods (before and after movement) by a co-registration of the two images. Backscatter values provide information on terrain roughness and any change indicates that a mass movement or a significant erosion event occurred in a given area. Vickers et al. (2016) conducted one of the first studies utilizing Sentinel-1 products to detect avalanches debris by developing an unsupervised classification. This technology seems very promising for avalanche detection (Eckerstorfer et al., 2017; Malnes et al., 2015; Martinez-Vazquez and Fortuny-Guasch, 2008; Schaffhauser et al., 2008; Tompkin and Leinss, 2021; Yang et al., 2020). Using TerraSAR-X and Sentinel-1 products, Leinss et al. (2020) mapped avalanches, demonstrating the potential of radar products in snow hazard

detection. However, the acquisition of frequent radar images is too recent to use this technique to detect historical avalanches. In addition to optical, radar or Lidar data, other studies used Digital Elevation Models (DEMs) and topographic parameters to determine the influence of terrain on avalanches in Switzerland (Maggioni and Gruber, 2009). Other studies incorporated other parameters such as morphology and vegetation to define potential avalanche zones and ran the Avalanche Flow and Run-out Algorithm to automatically detect potentially affected regions by avalanches (Barbolini et al., 2011). Moreover, the combination of snow measurements (depth) and high resolution DEMs have proved useful in snow hazard detection (Bühler et al., 2018a, 2022). Lidar is being used in the same regard with a higher level of precision. Lidar sensors measure snow depth before and after events at submeter resolutions (Prokop, 2008; Deems et al., 2013; Prokop et al., 2013; Hammond et al., 2018). However, this technology remains very expensive, and the spatial coverage is limited. Therefore, Lidar data are not suitable for avalanche detection at the basin scale. In addition to Optical, Radar or Lidar data, other studies used Digital Elevation Models and topographic parameters to determine the terrain influence on avalanches in Switzerland (Maggioni and Gruber, 2009). Some other studies added other parameters such as morphology and vegetation to define potential avalanche zones and ran Avalanche Flow and Run-out Algorithm to automatically potentially affected regions by avalanches (Barbolini et al., 2011). Moreover, the combination of snow measurements (depth) and high resolution DEMs proved its efficiency in snow hazards detection (Bühler et al., 2018a).

Optical data are the most available data in terms of spatial and temporal resolution as well as historical archives. Thus, we used optical data to detect avalanches on a long-term basis. Landsat-5, 7 and 8 products were used as their resolution (30 m, 900 m²) is sufficient to detect small avalanches (Eckerstorfer et al., 2016). Most of these data are available at a global scale. Optical sensors can detect areas covered or not covered by snow and this approach has been used in multiple studies during the past decade. Manual approaches or indices have been used in such studies. For example, Landsat-8 Panchromatic images (15 m) in combination with radar images were used to detect avalanches in Norway (Eckerstorfer et al., 2014). Such combinations were also recently used in west Greenland to map a large number of avalanches after an unprecedented snow event (Abermann et al., 2019). To our knowledge, only one recent study automated the detection of avalanches using remote sensing products and an open-access scripting approach (Smith et al., 2020). This study downloaded avalanches annually for a given region of interest using available Landast-8 images and computed NDSI for each image. NDSI differentiated so called 'supraglacial debris' from snow cover, for the date of interest. However, this approach only covers high elevations areas while our study aims to detect avalanches proximate to local communities at lower elevations (typically valleys). Manual and visual approaches, despite the time consuming process, can also be applied to detect avalanches using high resolution images (e.g., SPOT-6), mid-resolution (e.g., Sentinel-2A and B images), or even Google Earth images (Singh et al., 2020; Yariyan et al., 2020; E. D. Hafner et al., 2021). Across a wide area

(12,500 km$^2$) individual snow avalanches were manually digitized using high resolution SPOT-6 images (Bühler
et al., 2019a). Terrain parameters like slope gradient and curvature have also been added to the avalanche detection
process using DEMs combined with Landsat-8 images (Bühler et al., 2018b; Singh et al., 2019). Integrated criteria
are therefore recommended to detect avalanches. To our knowledge, no long-term avalanches mapping studies
using remote sensing have been conducted in the world, especially not in Afghanistan.

The general objective of this study is to map annual depositional zones of avalanche occurrence over the past 32
years using Landsat image archives in Badakhshan region, Afghanistan. Such long-term monitoring is the first
attempt globally and enables us to map the frequency of depositional zones of avalanches that impact valley
communities. Thus, we used optical data to detect depositional zones on a long-term basis and built an open-access
script in Google Engine interface: *Snow Avalanche Frequency Estimation* (SAFE). Landsat-5, 7 and 8 products
were used as their resolution (30 m, i.e., minimum detectible size of 900 m²) is sufficient to detect larger avalanches
(Abermann et al., 2019; Eckerstorfer et al., 2014, 2016; E. D. Hafner et al., 2021; Singh et al., 2019, 2020; Smith
et al., 2020; Yariyan et al., 2020). Our objective is to automatically map annual depositional zones of avalanche
occurrence over the past 32 years using Landast-5, 7 and 8 image archives in the Amu Panj basin of Afghanistan.
SAFE is applicable in any high mountains of the world, such as Tien Shen, Himalaya, Hindu Kush, Karakoram or
Andes, but not restricted to these, where snow avalanches deposits can be detected every year by satellite images
for a long time before completely melting. These outputs are of keen interest to decision makers who can use this
automated process to map avalanche hazard in the future. The most vulnerable areas, villages and roads, were
mapped for improve future planning. In addition, this research enables the monitoring of depositional zones of
avalanche evolution over the past 32 years. Such analyses should strengthen local community resilience to snow
avalanches.

**2. Materials and methods**
*2.1 Study area*
The study covers the most mountainous region of Afghanistan – Badakhshan in the Amu Panj basin located in the
northeast portion of the country. Average elevation is 2761 m and mean slope gradient is 21%. This region spans
from Bamyan Province to the Hindu Kush range, up through the Wakhan corridor in the far east of Afghanistan.
The summit is Nowshak Peak at an elevation of 7492 m a.s.l. The western part of Amu Panj basin is rather flat
and not prone to avalanches. Annual precipitation is about 600 mm occurring mostly as snow between February
and May (Zhang et al., 2015). This terrain and precipitation characteristics lend Badakhshan very prone to
avalanches. The basin is large (28,580 km²), justifying automated avalanche detection to cover this area in a
reasonable amount of time using Google Engine. Despite the remoteness of this region, Badakhshan has a
population of 950,953 inhabitants (Islamic Republic of Afghanistan Governmental Website, 2021) distributed in
4154 villages, mainly in valleys. However, 35% of the villages in Badakhshan are located at elevations above 2000
m, increasing the vulnerability of these communities to avalanches.

*2.2 Landsat archives for snowpack analysis*
This analysis requires the integration of numerous data into a Google Engine Java script. Firstly, a mosaic of
different Landsat images is created every year in the Amu Panj basin. Depending on the year of interest, Landsat-
5 (https://developers.google.com/earth-engine/datasets/catalog/LANDSAT_LT05_C01_T1_SR), Landsat-7
(https://developers.google.com/earth-engine/datasets/catalog/LANDSAT_LE07_C01_T1_SR), or Landsat-8
(https://developers.google.com/earth-engine/datasets/catalog/LANDSAT_LC08_C01_T1_SR) images were
downloaded. Within a given year the same satellite images were used. Before 1990, coverage by Landsat-5 was
insufficient in this region of Afghanistan. Landsat images were directly downloaded from Google Engine Archives
under their *ImageCollection*. Depending on the availability of images and the year of interest, one satellite or
another was used (Figure 1).

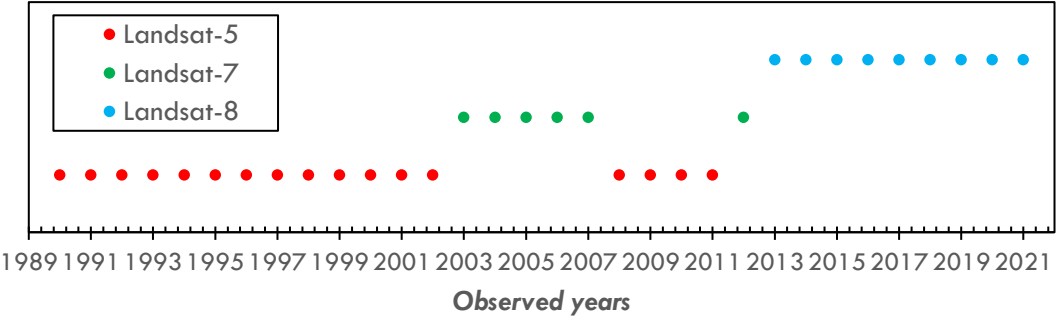

**Figure 1. Landsat archives used for depositional zones of avalanches detection since 1990.**

*2.3 Shuttle Radar Topography Mission-30 for terrain selection*
Detecting avalanche deposits requires terrain parameters defined by using the Shuttle Radar Topography Mission-
30 (https://dwtkns.com/srtm30m/). This Digital Elevation Model was collected in 2000 and is globally available
on the United States Geological Survey data portal at a spatial resolution of 30 m. SRTM-30 is used in this study
to delineate the regions of interest by deriving stream channels from the DEM.

*2.4 Terra MODIS MOD10A2.006 for snow line analysis*
The ROI (regions of interest) are delineated using Terra MODIS MOD10A2.006
(https://nsidc.org/data/MOD10A2/versions/6). This product of MODIS shows the snow cover (baseline: 8 days)
and is also globally available at a resolution of 500 m. MOD10A2.006 snow cover data are available since 2000.
MODIS is used to extract the seasonal snow line elevations (average) during the past 20 years in the Amu Panj
basin.

*2.4 Terra MODIS MOD11C3.006 for land surface temperature analysis*
The evolution of land surface temperature was completed using MOD11C3.006 monthly products, 0.05 degrees
(https://lpdaac.usgs.gov/products/mod11c3v006/). Temperature trends were analysed from 2000 through 2021
(significance > 0.05 p-value) and the slopes were extracted and plotted on monthly maps.

*2.6 Concept of the SAFE algorithm*
As the aim of this study is to detect and map the annual occurrence of depositional zones during the past 32 years
within the study area, the monitoring approach must be reasonable and transferable from year to year. Based on
frequent field observations and literature (Eckerstorfer et al., 2016), the authors noticed that depositional zones
can be detected using the contrast between snow cover and bare cover, but the timing is perhaps the most important
consideration. Indeed, the script is based on the assumption that snow packages exist in lowlands, especially along
rivers and streams, as late as May through mid-July. At this time of the year, the terrestrial snow cover has largely
melted and only snow packages triggered by avalanches remain. The location of those snow packages is also very
critical (i.e., along riverbanks). These zones are indeed detectable by delineating the depositional zones of the
avalanches (not their release or transition zones); in most cases these were located on river or stream banks as
observed in the field because the hillslopes always route snow avalanches in this direction. We cannot differentiate
between dry, wet, or powder snow because the process detects the remaining snow packages as avalanches in the
late season (spring and summer), not in winter, nor can we delineate multiple deposits within the same depositional
feature, only the combined deposit zones. In winter, we were not able to differentiate contrasts between snow cover
and avalanches, thus our focus was on the late season.

*2.7 Google Engine interface and code availability*
The concept of detecting the 'remaining snow avalanches deposits in the late season' was written in Java Script
using the *Google Engine* platform. The script SAFE is available at
https://code.earthengine.google.com/4653677dcdc6e02d4f8dfeca2fbf670f. We selected *Google Engine* for its
relative simplicity of use and open access code, which is available to all stakeholders involved in hazard and
vulnerability assessments. Additionally, internet connections in remote areas, such as within the Amu Panj basin,
are limited and powerful computers required to run scripts and process big data are sparse. Our script can be run
by anyone in a reasonable amount of time, even with a low internet capacity. As an example, yearly depositional
zones of avalanches in our study area were downloaded and mapped from Badakhshan (SAFE was processed from
Khorog, University of Central Asia campus, in Tajikistan) in 11.3 h (about 20 min per year of record) with an
average connection of 2.2 Gb/s.
*2.8 Region of interest*
The first step of SAFE is to define a region of interest as a mask to clip the Landsat images using SRTM-30 and
MOD10A2.006. Avalanche deposits that terminated on riverbanks, rivers, and streams are derived from SRTM-
30 DEMs using *ArcHydroTool* in *ArcGIS Pro Software*. Buffers of 200 m on both sides of rivers and streams are
defined to: (1) catch the depositional zone of avalanches that terminate in rivers and (2) increase the probability of
excluding the snow coverage in higher elevations that may remain throughout the summer. As an illustration, a
major avalanche occurred in the border zone of Afghanistan and Tajikistan in winter 2021. The remaining
depositional zone was still distinct in late May and June of that year on the bank of Chordara River (Figure 2).

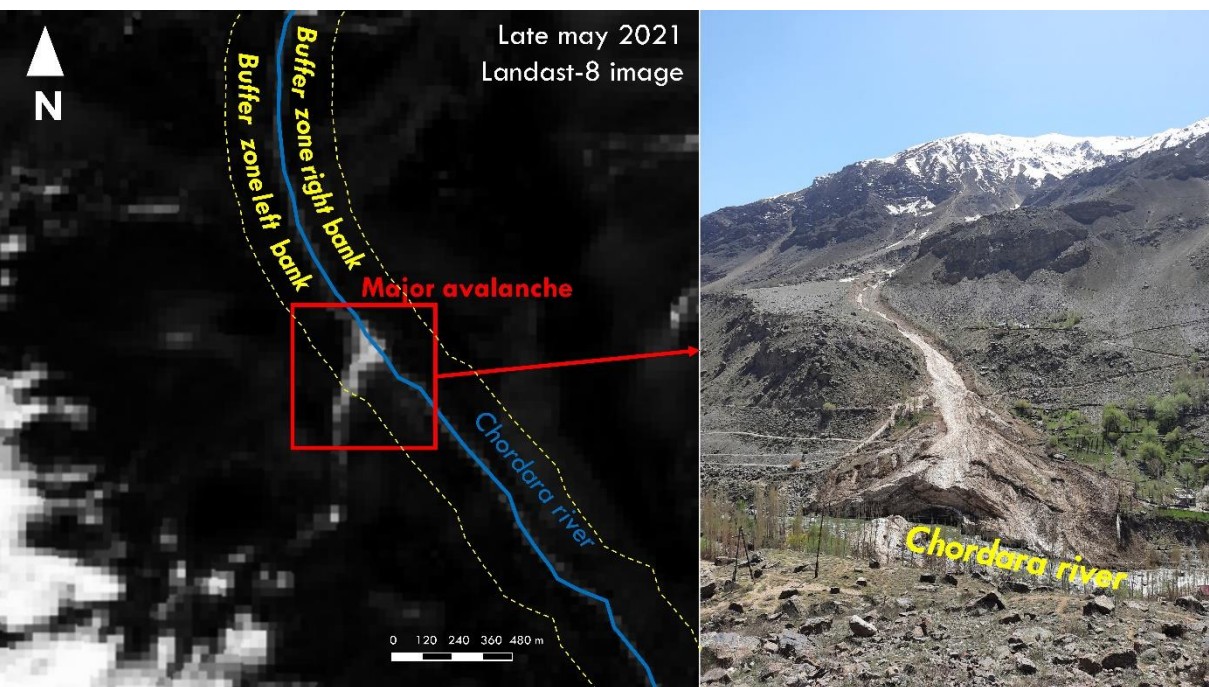

**Figure 2. An illustration of avalanche depositional zone detection using late snow season Landsat-8 image**
**near Khorog in May 2021 (Badakhshan in Tajikistan)**
*2.9 Date range of interest*
A 200 m riparian buffer was used as a mask to clip the Landsat images. Because our area of coverage encompasses
very different elevations, the date of snow melt is not uniform throughout the basin. Therefore, distinguishing
between the depositional zone and bare land requires different times depending on elevation. To accomplish this,
we calculated the average elevation of the snowline for the last 20 years using MODIS products. To distinguish
the different melt timing between highlands from lower areas, we selected the summer snowline (June-July-
August; JJA). The average elevation of the JJA snowline was 4420 m during the past 20 years. Two masks were
therefore produced: one with a river buffer in lowlands and another for highlands. Those masks are only relevant
if the user carefully selects the date of interest. For lowlands (below 4420 m), our time window was 15 May to 15
June, indicating that the script downloads and compiles all available Landsat images acquired in this range and
detects the deposit zones efficiently because during that period the terrestrial snow cover has already melted and
the deposits are easily recognised. For higher elevations (above 4420 m), snow cover melted later; dates to
accurately distinguish the remaining snow packages ranged from 15 June to 15 July. After many tests, it was
confirmed that these date ranges reproduced the desired snow conditions during the entire 32-y period. In the
script, users can modify these dates (line 24 and 112) to conform to local conditions.

*2.10 Snow index reclassification*
After the construction of the mask, SAFE proceeds as outlined in Figure 3. NDSI is selected to detect snow of
deposit zones in the script for its transferability from one Landsat generation to another. NDSI computes a ratio
between VIS and SWIR bands of Landsat satellites with negative NDSI representing non-snow cover and positive
values indicating snow coverage (Equation 1). Three cover types were distinguished to detect depositional zones
of avalanches at the correct time: (1) bare lands; (2) water bodies; and (3) snow. The values in Table 1 were
established after multiple tests before obtaining sufficient precision to distinguish deposit zones from other land
covers. On each mosaic (composite of the available images during the period of interest), a cloud mask is applied
using Landsat QA bands in the script to remove clouds from the scene.
$$\frac{Band\ 4 - Band\ 6}{Band\ 4 + Band\ 6}$$     **Equation 1**

**Table 1. NDSI discrete values for avalanche depositional zones detection**

| Coverage | NDSI values |
|----------|-------------|
| Bare soil | *-1 to -0.05* |
| Water bodies | *-0.051 to 0.30* |
| Snow cover | *0.31 to 1* |


*2.11 Depositional zone selection*
This further step reclassifies annual NDSI layers using ranges of values in Table 1. Only 'snow cover' that
designates snow avalanche deposit zones is selected in the script. From the selected reclassification, the script
removes the standalone pixels because their classification might not be precise or representative of actual cover.
Next, the selected 'avalanche pixels' are verified into the script avoiding manual vectorization after the
downloading process. The vectorization procedure of depositional zones of avalanches is justified by the analysis
steps and post-processing after downloading data. Depositional zones of avalanche statistics, elevations, and
surface areas are extracted from vector files. Finally, annual avalanche deposit zone shapefiles are exported into
the Google Drive user's account.

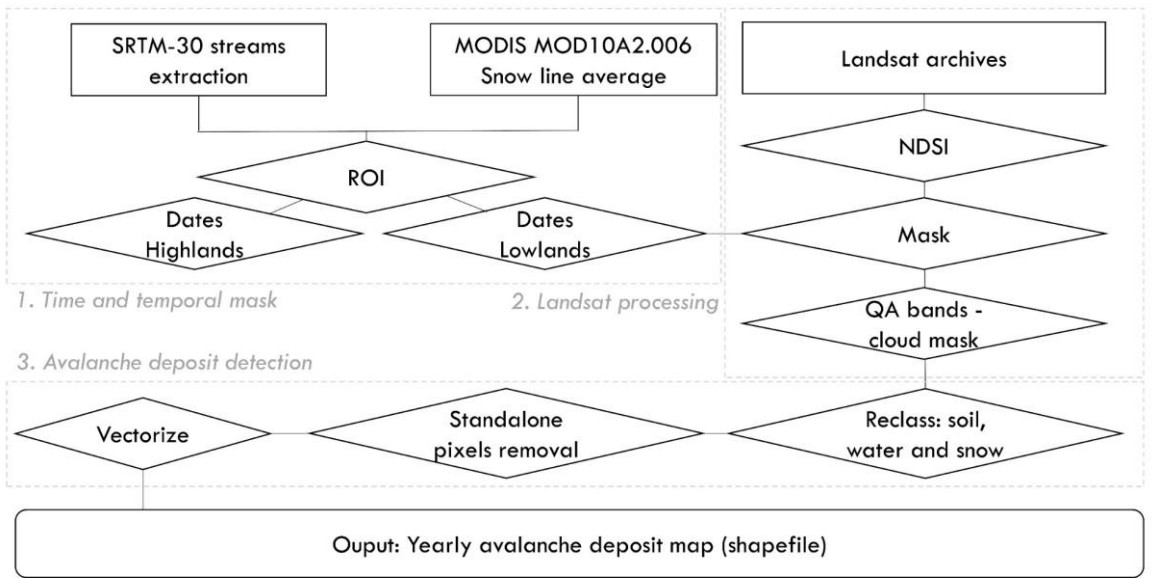

**Figure 3. Flow chart for Snow Avalanche Frequency Estimation (SAFE) using Landsat archives in Google**
**Engine.**

*2.12 Depositional zones of avalanche surface area classification*
Once the data are downloaded and imported into the GIS environment, statistical analysis commences. Every year,
the number and areas of deposit zones are calculated to quantify their evolution. Moreover, the surface areas of
the depositional zones are classified. Although a generic surface area classification exists (Greene et al., 2016),
we decided to classify avalanches by surface areas based on local conditions. We segregated four discrete
categories of deposit zones: small (< 1000 m²); medium (1000-5000 m$^2$), large (5000-15,000 m$^2$), and very large
(15,000-100,000 m$^2$). Such a classification enabled us to assess the intensity and potential impact of those hazards
in specific locations. SAFE is not able to detect the avalanches at their time occurrence, and since these hazards
are detected weeks after initiation, their surface area is underestimated by SAFE due to melting. However, the
estimated surface areas in SAFE are still useful for classifying depositional zones of avalanches by surface area
since large snow deposits melt slower than small snow deposits. The small avalanches that occurred in winter will
appear as small deposit at the time of extraction in SAFE and the large events as large hazards since visible snow
deposits can be seen in late spring.

**3. Results**
*3.1 Validation*
The performance of SAFE in correctly detecting snow avalanche depositional zones required careful assessment.
To achieve this, we collected datasets that show actual locations (Global Positioning System) of avalanches that
occurred in the Amu Panj basin during the last 32 years. A total of 158 snow avalanche depositional zones were
easily identified in the riparian buffer zones on Google Earth images in 2001, 2003, 2015, 2017, and 2019. No
other Google Earth images were available during the last 32 years in Afghanistan, therefore the comparison
between SAFE and the true events was conducted with those available 158 deposit zones. These 158 deposits were
extracted from Google Earth and stacked with SAFE outputs. SAFE deposits were considered as valid when the
two datasets were overlapped at the same location and when more than half of the polygons surface extracted from
SAFE was overlapping the actual deposits visible on Google Earth images. Here we used statistical measures to
assess the performance of SAFE through the Probability of Detection (POD; Equation 2, based on (E. D. Hafner
et al., 2021)):

$POD =$ true positive deposit zones/ (true positive deposit zones $+$ false negative deposit zones)
**Equation 2**

where *true positive deposit zones* are the avalanches detected by SAFE that were actually visible on Google Earth
images (in valleys were GE images were available) and *false negative deposit zones* are the locations where SAFE
did not detect deposit zones that had actually happened. Moreover, Positive Predictive Value (PPV; Equation 3)
was calculated to assess the number of times SAFE found an actual avalanche deposit zone on the ground as
follows:

$PPV =$ true positive deposit zones / (true positive deposit zones $+$ false positive deposit zones )
**Equation 3**

where *false positive deposit zones* are avalanche deposits predicted by SAFE that had never occurred.

The results suggest a good reliability of SAFE (Table 2). The overall POD is 0.77 which means that SAFE identified a significant number of the depositional zones of avalanches that impacted valley bottoms. Moreover, it seems that SAFE performs better in detecting true positive deposit zones (that occurred on the ground), as shown by the high PPV scores (average: 0.96). SAFE almost never detected depositional zones of avalanches that did not exist. However, SAFE might miss some deposit zones due to cloud cover on the Landsat images, especially in 2001 (Table 2; POD = 0.42 in 2001).

**Table 2. Probability of detection and Positive Predictive Values of SAFE**

| Statistics | 2001 | 2003 | 2015 | 2017 | 2019 | Average |
|---|---|---|---|---|---|---|
| **True positive** | 10 | 35 | 12 | 19 | 48 | |
| **False negative** | 14 | 6 | 1 | 4 | 9 | |
| **False positive** | 1 | 0 | 0 | 1 | 3 | |
| **POD** | 0.42 | 0.85 | 0.92 | 0.83 | 0.84 | **0.77** |
| **PPV** | 0.91 | 1.00 | 1.00 | 0.95 | 0.94 | **0.96** |

Another source of error arises when SAFE cannot detect depositional zones due to a dark color on the snow surface associated with surface debris or a debris flow on top of the deposit zones. NDSI may have identified those debris layers as bare soil in the classification. Moreover, it should be understood by the users that another limitation is that SAFE does not detect early winter avalanches deposits due to melting and snow coverage on and around the snow deposit, which might affect the deposits frequency estimations. However, based on our findings, SAFE can be considered as a conservative, yet robust and efficient tool to automatically identify snow avalanche depositional zones in very remote areas and can be applied in any mountainous region.

*3.2 SAFE outputs compared with outlined avalanches using SPOT-6 images*

As a potential method of strengthening our testing of SAFE, outputs of our model were compared with a method that applied a more precise and expensive remote sensing product in Switzerland in 2018 (Bühler et al., 2019b; Hafner and Bühler, 2018). The Swiss area encompassed 12,500 km² where more than 18,000 snow avalanches were manually digitized using very high-resolution products SPOT-6 images (in January 2018). While our dataset is quite different from the Swiss data, the objective of this comparison was to assess how many snow avalanche deposits SAFE could detect compared to the approach using SPOT-6 (Table 3). Figure 4 shows an illustration of this comparison. It appears that the deposit zones detected by SAFE are in line with SPOT6 outlined avalanches. The later however covers the entire avalanches while SAFE only detects, automatically, the deposit zones.

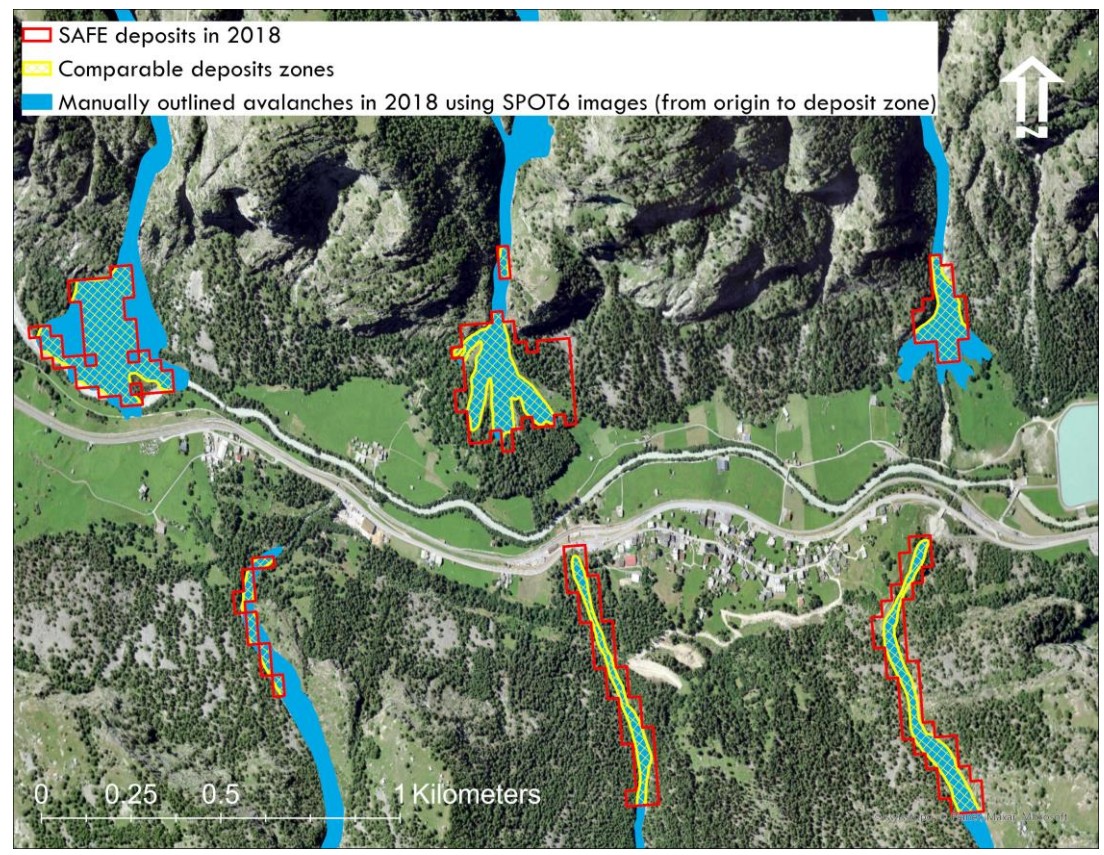


**Figure 4. An illustration of the comparison between automatic detection of deposit zones using Landsat**
**archives in SAFE and manually outlined snow avalanches (from origin to deposit zones) using SPOT6**
**images in Switzerland**

**Table 3. Comparison of snow avalanches deposits zones between SAFE outputs (April to June 2018) and**
**manual digitization using high-resolution SPOT-6 images in Switzerland in January 2018\***

| Method | Number of snow polygons | Area of snow polygons (m²) |
|---|---|---|
| SPOT digitization | 7574 | 362,187,741 |
| SAFE detection | 9948 | 494,454,599 |
| Overlapping SPOT-SAFE | 2194 | 223,907,868 |

*SPOT data based on (Bühler et al., 2019b; Hafner and Bühler, 2018)

Importantly, not all avalanches manually digitized on SPOT-6 images were comparable to SAFE results. To make
this comparison more consistent, we clipped the outlined avalanches with the valley bottom mask used in SAFE.
Following this modification, the SPOT-6 digitization process identified 7574 avalanches deposits in valley bottoms

compared with 9948 by SAFE. Overlapping these two datasets, we found that both approaches detected 2194 deposit zones in common. Much of this discrepancy is due to the timing of SAFE images, which examine deposits that remain in late spring and early summer, whereas SPOT digitization covered only January. The larger number of snow deposits detected by SAFE occur during late season snow avalanches that impact valleys. This suggests that SAFE could not detect all January snow deposits because many of those already melted by the time of SAFE detection (early April to late June in the Swiss case). In addition, optical image quality strongly depends on cloud cover that may cause avalanches to be obstructed. For instance, we could not compare the 2019 SPOT-6 derived dataset in eastern Switzerland (E. Hafner et al., 2021) due to cloudy images at the end of winter and early spring because these snow avalanches had already melted, implying that SAFE is more suitable for high mountain areas (>4000 m) where snow deposits remain longer in valleys, thus inflicting greater damages and obstructions. Using LANDSAT images, SAFE somewhat circumvents this problem of cloud cover by assessing many years of data (in our case 32 y). However, SAFE does not distinguish individual events and considers overlapping snow deposits as one, in contrast to SPOT-6 which distinguishes these as discrete events. This, in addition to the different methods and spatial resolution difference between SAFE and SPOT, explains the somewhat low number of overlapping snow deposits between SAFE and SPOT. Moreover, the SPOT digitization procedure found a total avalanche area of 362,187,741 m² in January, while SAFE detected 494,454,599 m² of deposits at the end of the avalanche season, including 223,907,868 m² in common. The area detected by SAFE is naturally larger than SPOT-6 since SAFE maps all detectable deposits at the end of the winter. Moreover, SAFE did not detect the small avalanches of January that rapidly melted after they occurred. The polygons extracted by SAFE using Landsat images are obviously coarser than those outlined with SPOT-6 images, which partly explains the low number of overlapping snow deposit zones, but a much more comparable detected area (62%) between the two methods. Much of the discrepancy is related to SAFE's inability to detect individual events and missing deposits that rapidly melt (mostly from the early winter snow avalanches), as well as the very different resolution of these products.

*3.3 Snow avalanche depositional zone frequency from 1990 to 2021*

By compiling 32 years of satellite images (see Methodology), the frequency of avalanche depositional zones at a 900 m² pixel scale was determined (Figure 5 and 6a). SAFE inventories snow avalanche deposits that occurred within a year and therefore identifies the most vulnerable areas, but it does not aim to forecast future avalanches. During this period, some 810,000 depositional zones impacted valleys within the Amu Panj basin (28,500 km²), i.e., approximately 28 depositional zones km$^{-2}$. Each year these avalanche deposits cover an average of 1.23% of the basin area but surface area varies from year to year. Avalanche depositional zone surface area ranged from 900 to 100,000 m$^2$ and is categorized into four classes: small (< 1000 m²); medium (1000-5000 m$^2$), large (5000-15,000 m$^2$), and very large (15,000-100,000 m$^2$). The most frequent are medium surface area deposit zones; 342,000

events during the past 32 years. Our approach also identifies very large snow avalanches depositional zones that
pose the greatest danger to local populations and infrastructure. We found no correlation between altitude of
depositional zones and their surface areas. Avalanches deposits in this region have an average altitude of 3820 m
and the lowest depositional zone occurred at 1755 m.
These spatial and temporal statistics allow for a geographic assessment of the avalanche deposits. In total, ten sub-
catchments (ranging from 18 to 240 $km^2$) were impacted by more than one avalanche depositional zone $km^{-2}y^{-1}$,
with an average frequency of 0.26 deposit zones $km^{-2}y^{-1}$ throughout the Panj Amu basin (Figure 7). More
importantly, these maps prioritize villages prone to avalanches deposits and inform relevant stakeholders which
villages and infrastructure are most at risk. Of the 4154 villages in the region, 50 are impacted by at least one
avalanche within a 1 km radius each year (Figure 8). These susceptible villages are in Upper Badakhshan in the
north of our study area and in the Wakhan Corridor in the east where the highest mountains and most remote
villages are located. During the 32-y period, 92 villages were affected by very large avalanches depositional zones
in Badakhshan and Wakhan. Since 2019, Aga Khan Agency for Habitat (AKAH) is monitoring villages of
Afghanistan that have been impacted by snow avalanches. In total, 217 villages have been impacted by avalanches
deposit zones and those are located in the same vulnerable valleys detected by SAFE, namely High Badakhashan
and the Wakhan corridor.
Our remote sensing approach facilitates innovation in snow avalanche depositional zone monitoring: i.e., detecting
avalanches deposits outside of populated areas, especially along roads that are frequently blocked by avalanches
(Figure 9). More than 2000 roads in the basin (5.47% of the road network) were affected by avalanches deposits
every year. Additionally, more than 400 roads in Upper Badakhshan and Wakhan regions experienced more than
2 avalanches depositional zones $y^{-1}km^{-1}$ of road (within a 1 km buffer). The average frequency along roads is 0.86
avalanche deposits $km^{-1}y^{-1}$ during the past 32 years, most of these in the medium- surface areas category.

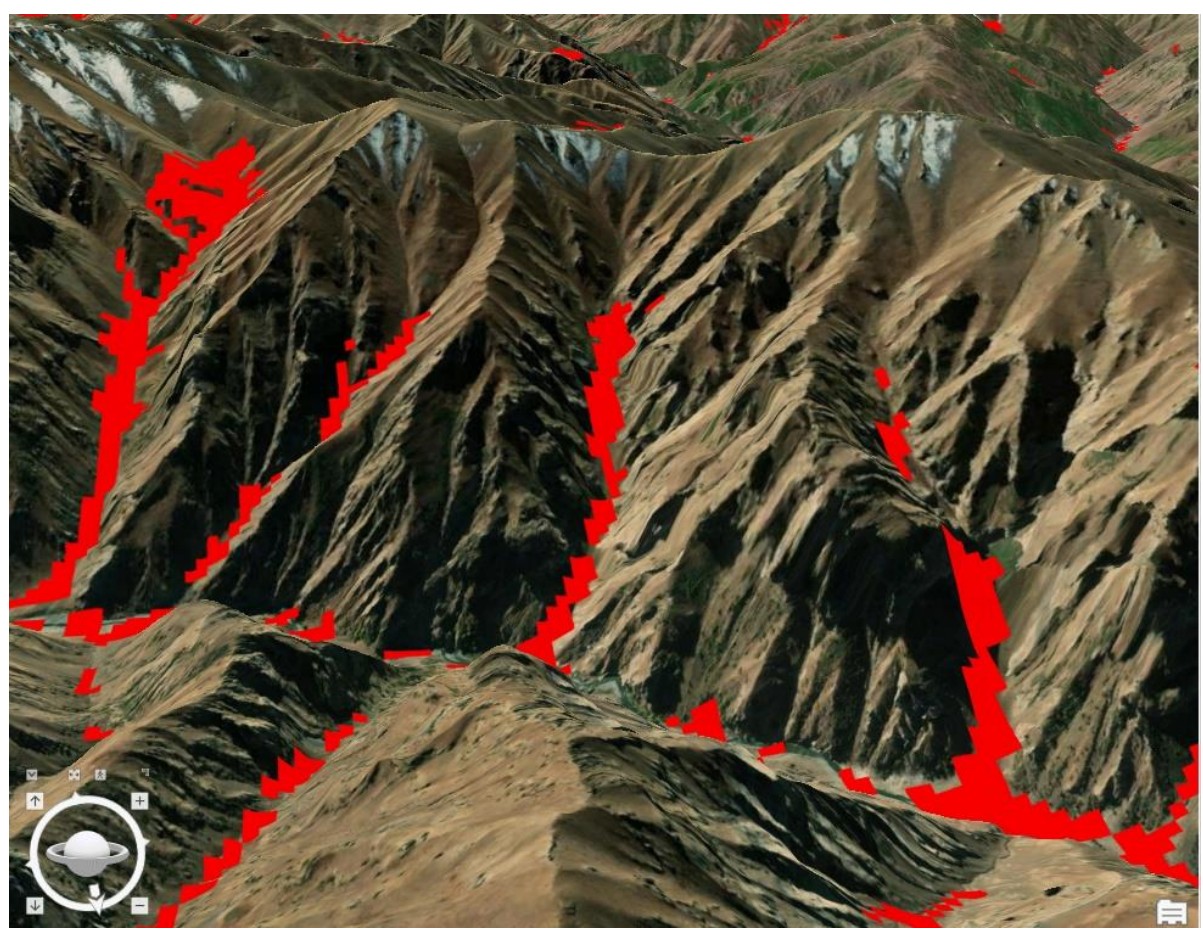


**Figure 5. 3-Dimension view of the 32 years avalanche depositional zones maps in Khinj village in**
**Afghanistan (*ArcGisPro*)**

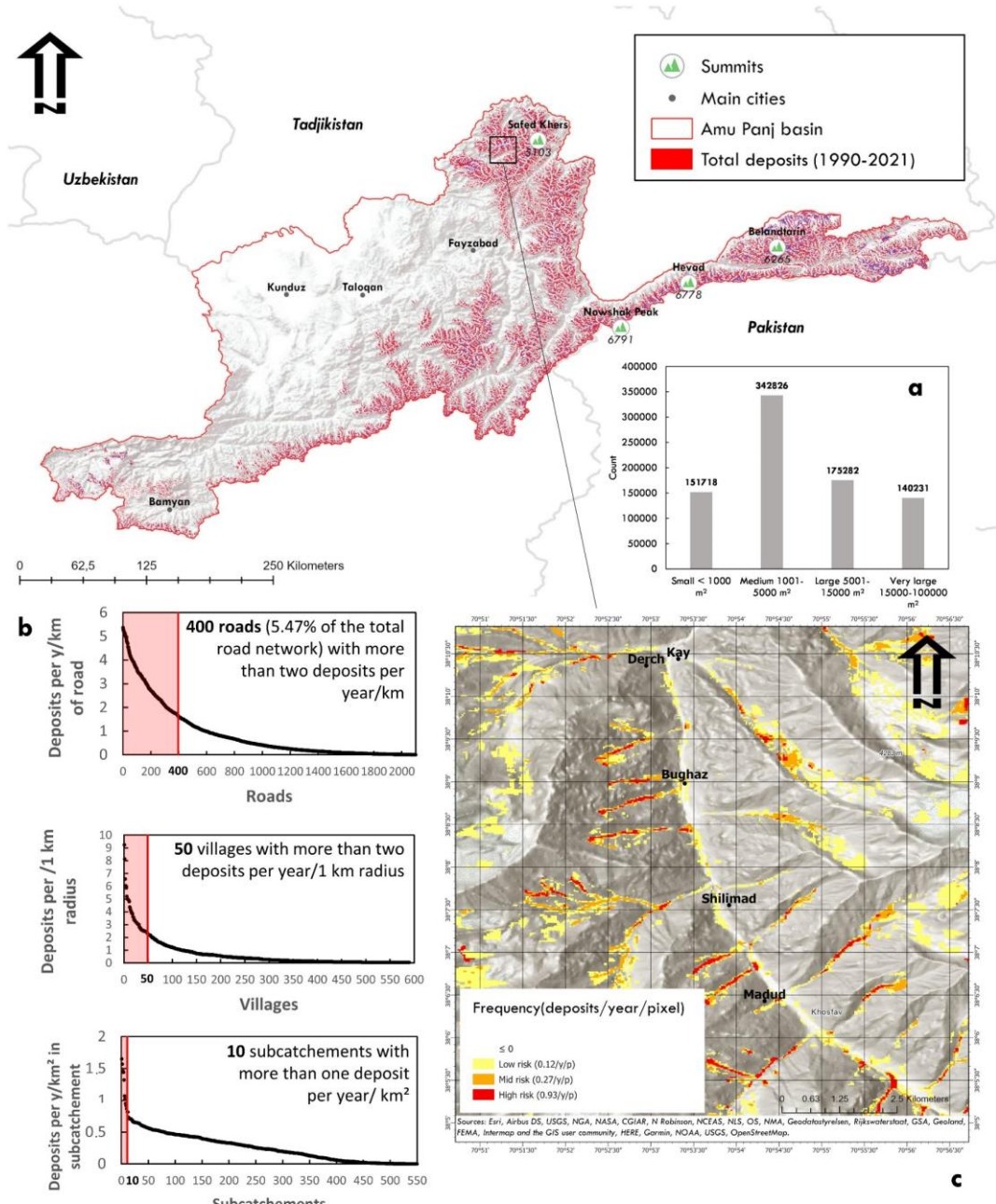

Figure 6. Yearly inventory map of snow avalanche depositional zones in the Amu Panj basin: 1990-2021: a, Surface area classification of avalanche depositional zone frequency; b, Avalanche depositional zone frequency per number of roads, villages, and subcatchments in the basin; c, An example map of avalanche depositional zone frequency during the 32-y period at a village scale.

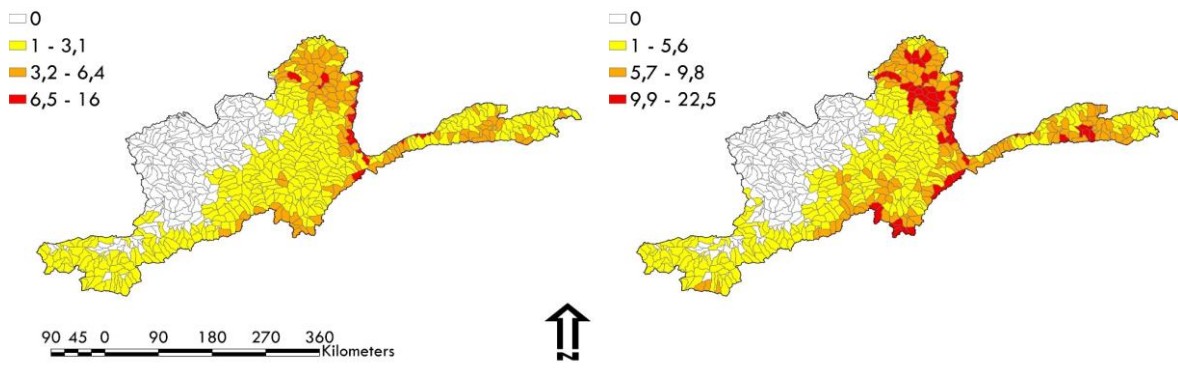


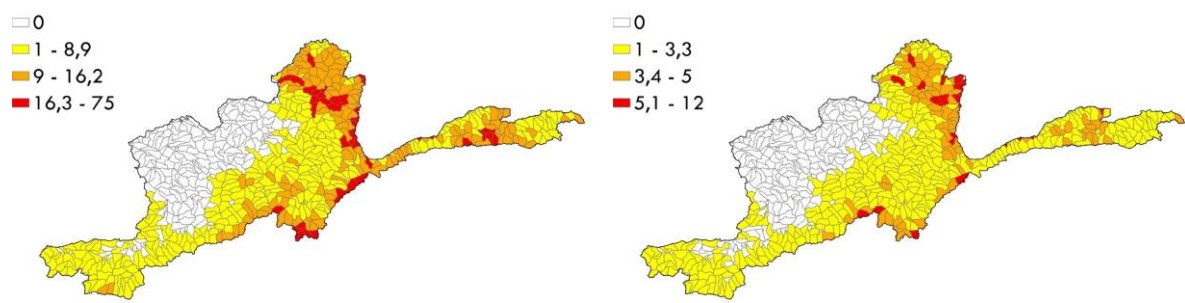


**Figure 7. Total avalanche depositional zones per category and per square kilometer in subcatchments**
**during the past 32 years.**

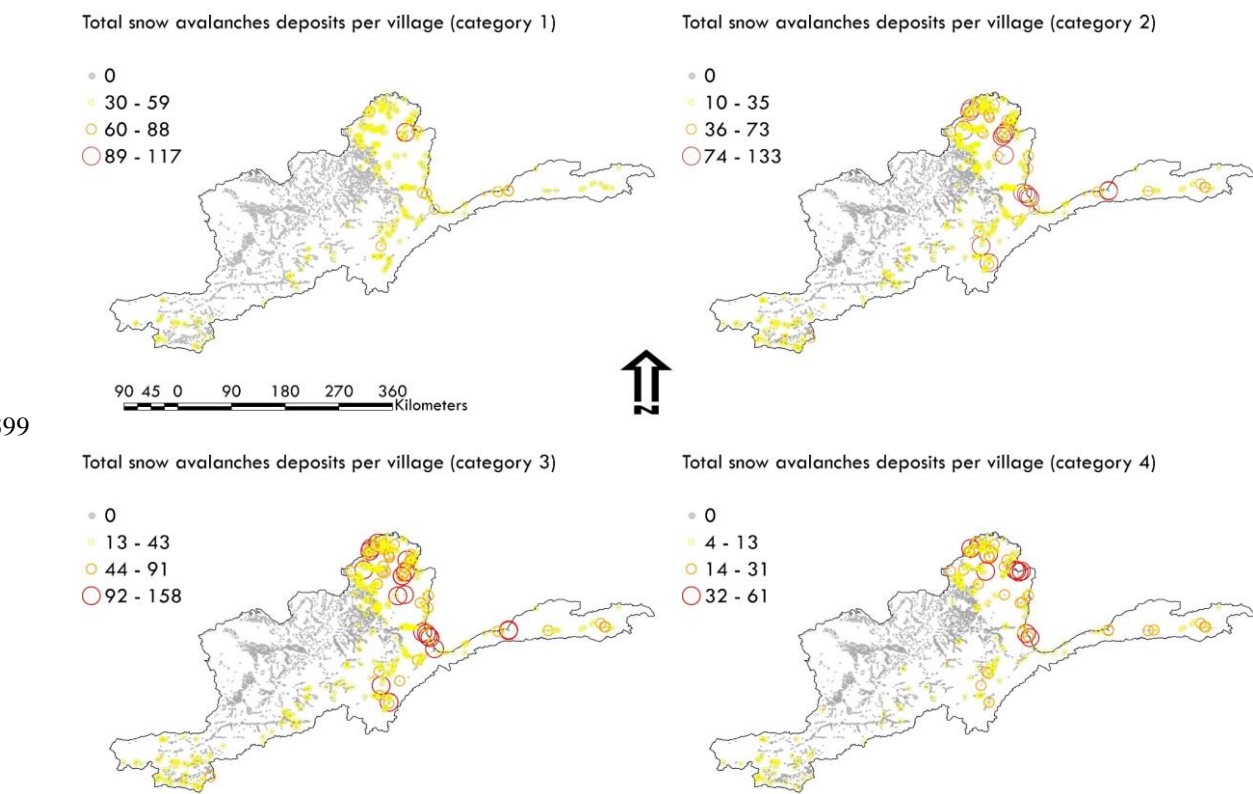


**Figure 8. Total avalanche depositional zones per category and per village during the past 32 years.**

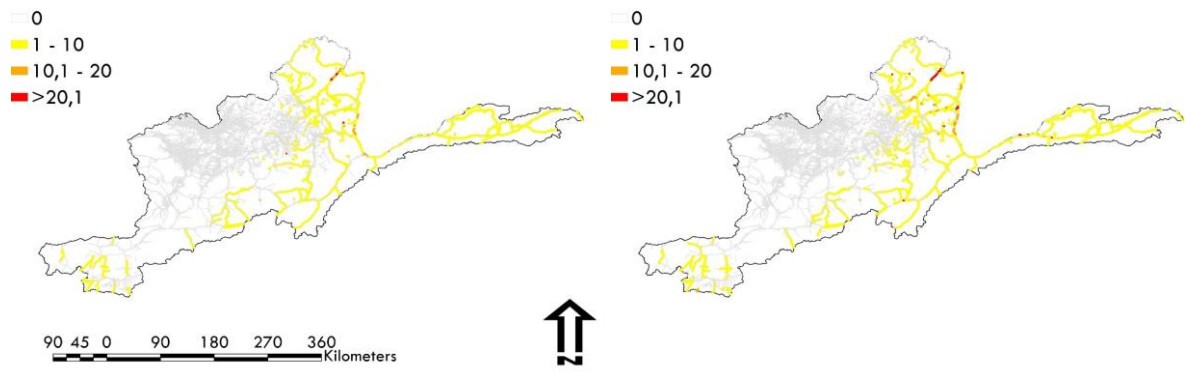

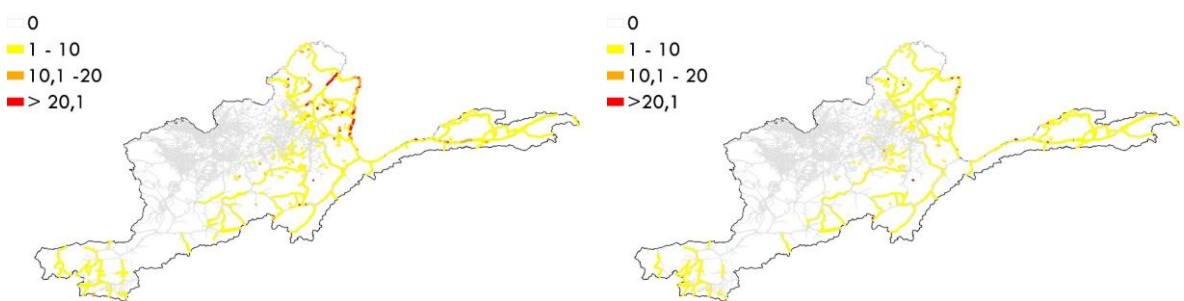



**Figure 9. Total avalanche depositional zones per category and per kilometer of roads during the past 32**

**years**





### 3.4 Stream blocking and resultant flooding

Damages to infrastructure and blocking of roads by avalanche deposits are not the only consequences of these mountain hazards. Because depositional zones typically reach rivers in this steep, incised terrain, the sudden and rapid arrival of several tons of snow can temporarily block rivers inducing short-term localized flooding. By cross-checking the map of the rivers in the Amu Panj basin with SAFE outputs, it appears that 26.2% of the river network is impacted by avalanche deposits, mainly in the high mountains. During the past 32 y, 12% of the streams have been blocked by at least 10 avalanches deposits km$^{-1}$ representing a significant risk for villages and farms in floodplains. The accumulated snow mass impounds river water until it can break through releasing a large discharge surge. Thus, depending on the surface area of the avalanche deposit with respect to the channel dimensions, damages to villages and farmlands may occur both upstream due to impounded water (hours to weeks) and to downstream following the sudden release of water.

### 3.5 Snow avalanche depositional zone trends during the past 32 years

This long-term monitoring of snow avalanche deposits facilitates the assessment of the evolution of these rapid mass movements. During the 32 years of avalanche depositional zone assessment, no significant temporal trends in impacted areas were detected (Figure 10). In addition, there was no significant trend of the surface are of snow deposits (p-value > 0.05). Nevertheless, some years posed much greater risk than others. In the last 32 years, ten years have been more at risk with above-average avalanche deposit coverage: 1990, 1991, 1992, 1993, 1994, 1995, 1996, 2003, 2005, 2007 and 2012. In particular, 2003 had many avalanche depositional zones that occupied almost 6% of the surface area of the entire basin. That year was locally noted as having heavy snowfall and farmers benefited from more snowmelt in the spring, leading to higher than average crop yields in 2003 (FAO, 2003; Guimbert, 2004). Notably, the higher risk years were also characterised by lower altitudes for avalanche deposits. There is a slight negative correlation (-0.55, Pearson test) between altitude and total annual avalanche area. With larger avalanche areas, deposits reach closer to villages. For example, in 2003, the lowest avalanche depositional zone occurred at an altitude of only 1871 m, very close to housing clusters and roads. It is therefore possible that communities below 2000 m are also impacted by snow avalanche deposits and in many mountain regions of the world this represents a significant proportion of the communities living proximate to these altitudes.

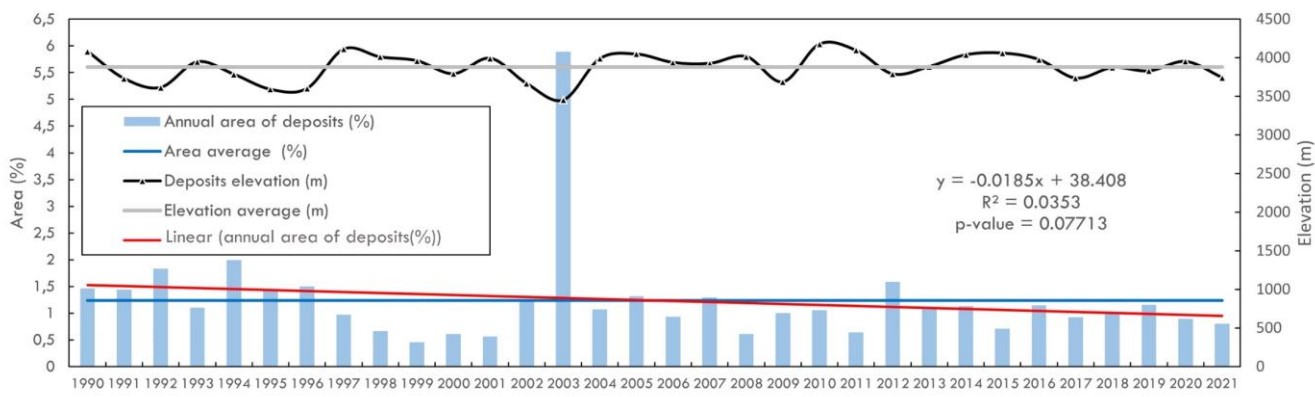

**Figure 10. Snow avalanche depositional zone area and elevation trends since 1990 in the Amu Panj basin. Elevation was calculated within each polygon of avalanche deposits using SRTM-30 Digital Elevation Model. Mann-Kendall p-value 0.05 test was conducted to assess the significance of the trend.**

*3.6 Temporal geographic shifts of snow avalanches deposit zones*

Long-term monitoring also shows the evolution of the spatial distribution of snow avalanche depositional zones. The pattern of snow avalanche deposits has changed with time and slightly shifted to the northeast portion of the basin; thus, more avalanches are now occurring in the northeast than in the southwest (Figure 11a and b). Nevertheless, snow coverage did not

shift simultaneously according to our remote sensing analysis nor did the snowline evolve, but rather remained variable over the last 32 years. The geographic shift of avalanche depositional zones is therefore likely due snow depth evolution. Deeper snowpacks trigger snow avalanches. There are no available data on snow depth at such a scale. However, the slope was calculated and a Mann-Kendal test was applied for each pixel of the land surface temperature images (MOD11C3). Remotely sensed land surface temperature changed during the last 20 years (Figure 11 a), with a warmer band occurring through the

central portion of the basin in December (p-value 0.03 with an increase of 0.88 C°y⁻¹). This central portion is mainly mountainous and this temperature pattern may have shifted the avalanches to the northern mountains of the area, while the south is characterized by lower mountains. Overall, avalanche depositional zone locations tend to follow the spatial distribution of snow depth (Bühler et al., 2016). This means that despite the high variability of the snow line and snow coverage, the distribution of snow avalanche deposits can significantly change over time in response to temperature changes and local

communities must be prepared for shifting hazards.

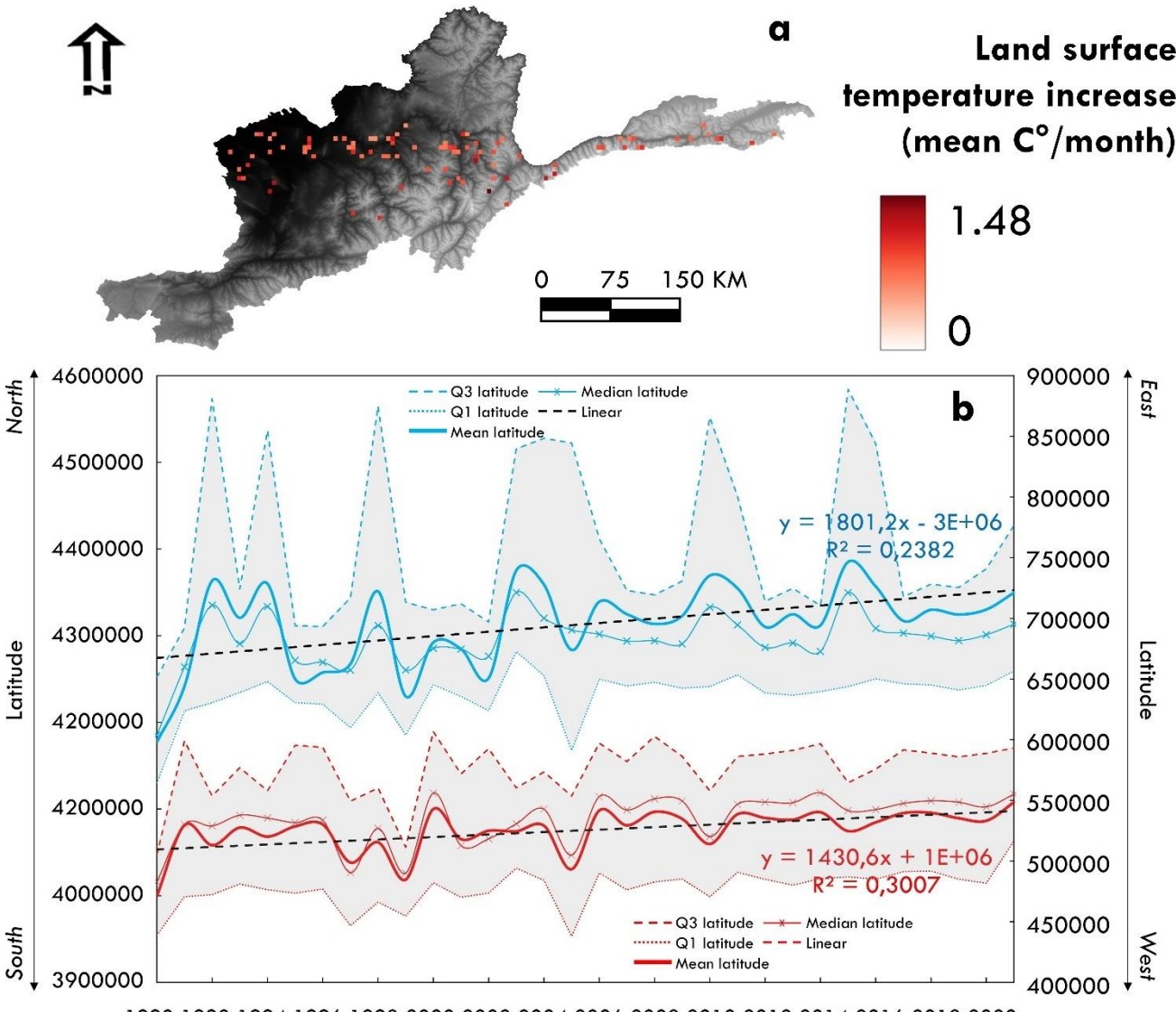

**Figure 11. a, Map of areas with significant increases in monthly land surface temperatures in the Amu Panj Basin based on MOD11C3 products from 2000 to 2021; b, Geographical shift of avalanche deposits: mean longitude and latitude of avalanche deposits each year since 1990 show evidence of a movement to the northeast due to increasing winter temperature in mountainous areas.**

## 4. Uncertainties and implications

*4.1 Sensitivity analysis of SAFE*

To better understand how SAFE works and assess its performance, a sensitivity analysis was conducted between the model parameters. The number and surface areas of avalanche depositional zones vary according to the buffer used, the dates of

Landsat images, and finally the NDSI range during the snow classification. The sensitivity analysis was conducted for the year 2019 when SAFE was most robust in valleys where actual avalanche deposits were quite visible on Google Earth images (POD: 0.84 and PPV: 0.94). First, we run SAFE with different buffer widths (25 m of difference between each buffer). There is a strong positive correlation (0.98) between the number of avalanche deposits detected by SAFE and the buffer width (Figure 12a). The wider the buffer around the rivers, the more avalanche deposits SAFE will detect. On the other hand, for narrower buffers, the average surface area of avalanche depositional zones is smaller (positive correlation of 0.71). This is because a large buffer extends upslope where small snow patches reside, which are not avalanches deposits since they are located at the top of hillslopes. This means that the user should not select a buffer that is too wide, rather the area should only include the riparian zone of rivers and streams where the snow avalanche deposits are located. As such, we used a value of 200 m for the entire region studied.

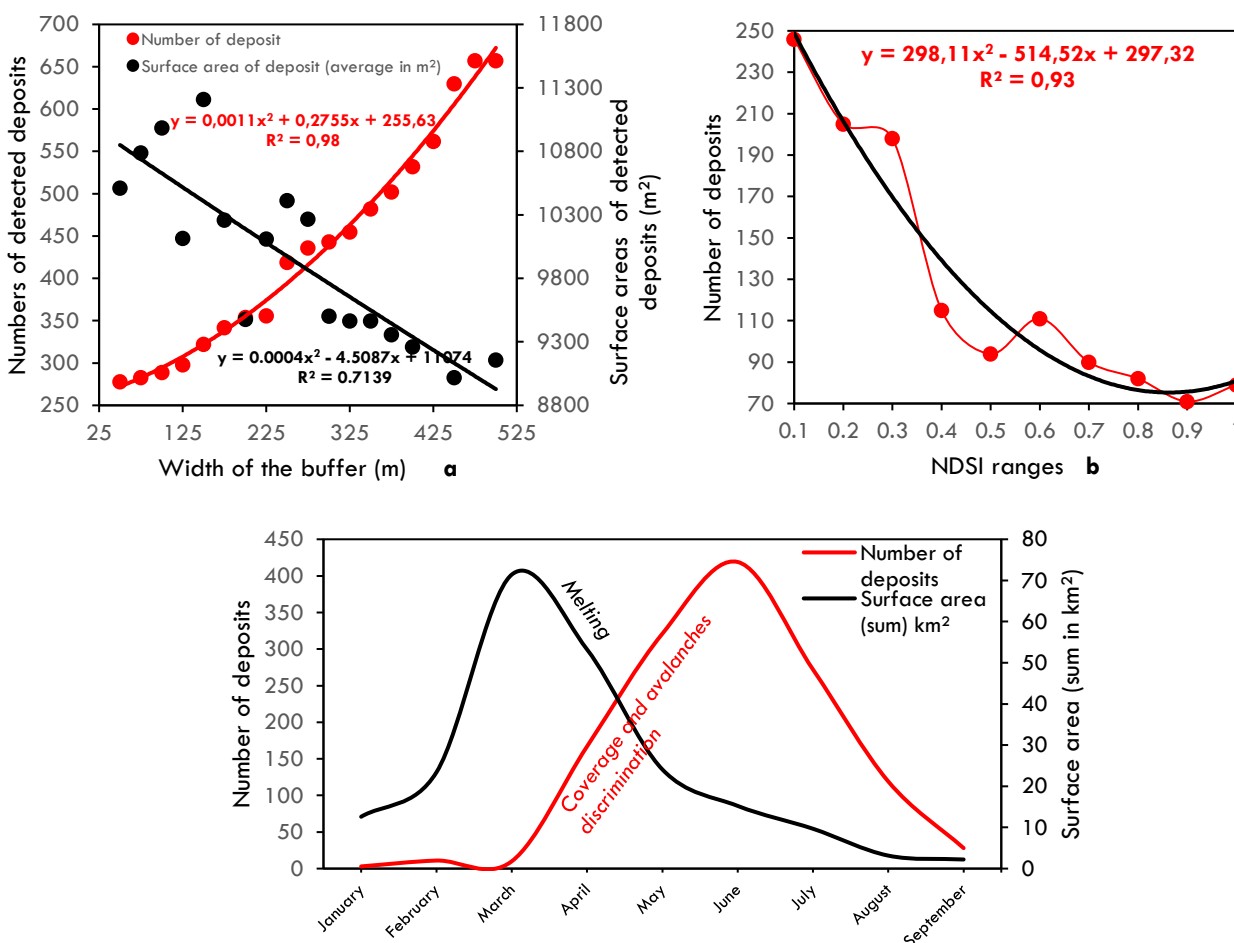

**Figure 12. Sensitivity tests of SAFE for number and surface areas of detected avalanche depositional zones: a, width of buffer; b, NDSI ranges; and c, dates of interest**

The number and surface areas of avalanche depositional zones detected by SAFE depends on the NDSI range when classifying snow. NDSI is used to differentiate between water bodies, bare lands, and snow. By varying the NDSI ranges of snow in the script, we notice a strong positive correlation with the number of avalanches deposits detected by SAFE. The closer the index is to 0, the more hazards SAFE finds. However, this correlation shows us that the choice of NDSI range is important because we notice a threshold at 0.31 (Figure 12b). Avalanche depositional zones seem to be more numerous with an NDSI lower than 0.31 because the snow pixels are confused with water bodies. It is therefore essential for the user to select an NDSI higher than 0.31 to distinguish water bodies (rivers, flood areas or lakes) and snow. However, there is no correlation between the NDSI ranges and the average surface areas of avalanche deposits because NDSI cannot interpret pixels other than 'snow' above the 0.31 threshold. Finally, the date of interest is a key parameter in SAFE. The number of avalanche depositional zones detected by SAFE is highest at the end of winter due to the almost constant cloud cover since January, but also due to the inability to distinguish avalanche deposits from snow cover in winter (with Landsat images) (Figure 12c). May is a key month for SAFE applications in high mountains: the snow coverage, which is thinner than the avalanche depositional zones, begins to melt and the number of avalanche deposits detected can then be assessed. It is therefore essential to select post-May images to detect avalanche deposits while taking care not to select post-July images as avalanches deposits melt in summer precluding detection.

Some avalanche deposits are also visible on successive images after the snow cover melts. SAFE was specifically designed to detect avalanche depositional zones at their earliest stage after snow melts. Indeed, starting from May (when the avalanche deposits are not confused with snow coverage), snow avalanche deposits start to melt and the surface area will begin to be underestimated. For that reason, it is important to select late spring images for lowland avalanche depositional zones and early summer for high land deposits, not later. Cloud cover is another issue in avalanche deposit locations and surface area detection since cloud cover may partly or fully obstruct the avalanche deposits at the time of the image. This is another reason to select images starting from late spring when regional cloud cover is lowest and even absent in early summer. If cloud cover is high even late spring, the users can still select later images, but there will be a risk that detected avalanche deposits will have started to melt. To summarize, we recommend the following three parameters in the SAFE script: buffer of 200 m to include only snow avalanche deposits; NDSI > 0.31 to distinguish water bodies from snow, and images from May to July to distinguish avalanches depositional zones from snow cover.

*4.2 Excluding snow coverage*

Interpreting the remaining snow packages as avalanche deposits can lead to some errors. Indeed, despite a precise masking operation (excluding summits and very high plateaus where snow persists), in some cases the use of NDSI might not properly segregate avalanche depositional zones from large areas of remaining snow. After assessing the surface area of true avalanche deposits (the ones that SAFE correctly detected based on Google Earth images), it appeared that snow cover > 100,000 m² were not avalanche depositional zones but rather snow cover; thus, these were removed. However, in highlands, even along

riverbanks, some snow packages interpreted as avalanche depositional zones may be remaining snow cover. As such, the date range for highlands was selected as late as possible in the year. Thus, it is advised to keep the mask at the very bottom of valleys (maximum 200 m buffer along the river) to exclude high plateaus and potential snow-covered areas.

*4.3 Water bodies in SAFE*

A final limitation of using this remote sensing and NDSI approach for avalanche deposit detection is the possible confusion between some small water bodies and avalanche depositional zones. Indeed, in some cases certain river reaches (stream order > 4 in our study area) could be interpreted as snow because they were frozen and appeared as white pixels on Landsat archives. The same issue can occur with ponds and lakes. This limitation was foreseen before processing the images in our study and we excluded these large water bodies from the region of interest (in the mask) by using available shapefiles. For example, Shiva Lake, one of the largest water bodies in Amu Panj basin (15 km²), was removed from the analysis. Another way to avoid the water pixel selection is to adapt the NDSI reclassification itself, depending on the study area. This is possible in the script lines 51-53 for low elevations and lines 139-141 for high elevations in the script.

*4.4 SAFE outcomes compared to other snow avalanches detection studies*

SAFE contributes to the literature on snow avalanche detection, but in a unique way using remote sensing. As noted, many studies and models exist using various products: Radar, Optical, and Topographic. The strength of remote sensing is the automatic processing at a large scale and over long timeframes. SAFE uses the capabilities of remote sensing by processing more than one image per year at the catchment scale. Moreover, the use of Landsat archives allows assessment over the last 32 years, which is not yet possible with recent Radar data such as Sentinel-1. Most of the current avalanche detection models use freely available products, with acceptable if not good accuracy (Table 4). The accuracy of these studies using Radar images ranges from 53 to 81% making this a relatively robust tool. One of the reasons why SAFE does not use Radar images is the weight of the images (data storage), especially Sentinel-1, which is mostly above 1 Gb/image. These heavy images are not suitable for a model like SAFE, which was specifically designed for remote study areas where internet connections may be very limited. Other models also exist with Optical images with high accuracy ranging from 71 to 93% (Table 4). In the optical domain, SAFE showed a POD of 77% over an area of 28,500 km². SAFE is therefore in the high range of models with optical, medium resolution (Landsat) images.

**Table 4. References on snow avalanche accuracy using remote sensing products (Radar, Optical and Terrain)**

| Reference | Accuracy (%) | Dataset |
|---|---|---|
| Eckerstorfer et al., 2017 | 75 | S1 |
| Malnes et al., 2015 | 53* | S1 |
| Martinez-Vazquez | 76 | GB-SAR LISA |
| Tompkin and Leinss, 2021 | 81 | S1 |
| Leinss et al., 2020 | 70 | S1 and TerraSAR-X |
| Vickers et al., 2016 | 60 | S1 |
| Karas et al., 2021 | 70 | S1 |
| Yang et al., 2020 | 75** | S1 |
| Singh et al., 2019 | 93 | L8 |
| Yarivan et al., 2020 | 90 | Google Earth imagery |
| Hafner et al., 2021 | 74** | SPOT |
| Bühler et al., 2018 | 95** | DTM |

*55 avalanches were detected using S1 image out of 102 on the field.

**POD

## 5. Conclusion

SAFE can be considered as a universal approach to assess snow avalanche depositional zones in spring and early summer
where ground data are very limited, such as in the Afghan mountains. Here we showed the capability of long-term remote sensing data to robustly detect snow avalanche deposits that impact valley locations. While we have successively applied SAFE to assess the frequency and impacts of avalanche deposits in valleys and lower hillslopes of Afghanistan, arguably one of the most data-limited regions worldwide, this model should perform even better in areas where snow data are available making it an important tool for avalanche vulnerability assessment worldwide. More than 30 years after the launch of Landsat-
5, it is now possible to compile all data and assess the temporal as well as spatial evolution of such hazards. NDSI is a relevant index to detect avalanches when selecting the correct region and dates of interest - i.e., riverbanks during the late melt season. The thickness of the depositional zones facilitates the detection of these avalanche deposits after the snow cover has melted on hillslopes in spring or early summer. Moreover, the application of SAFE in Afghanistan, compared to its application in Switzerland, showed that the script can be applied worldwide, especially in high mountains (above 4000 m) since deposit
zones are still detectable in late spring at those elevations.

The automation of snow avalanche detection using remote sensing technologies at regional scales is still new and SAFE was designed to guide decision-makers, planners, and disaster risk practitioners. Indeed, such people can now know where the most at-risk areas are located based on these frequency maps. Such information informs the relative risk of building sites and land use decisions in such mountainous terrain with greater precision. The level of exposure of roads to avalanche depositional
zones can also be estimated using these frequency maps, and can inform road planners and managers regarding road location,

maintenance practices, and mitigation structures. Moreover, villages of high mountains such as in Afghanistan are strongly highly dependent on roads connections to provide necessary food, energy, medical supplies, and life-support items, especially in winter. It is therefore critical for local decision makers to assess the frequency of road blockage by avalanche deposits. Thus, open-access and user-friendly tools such as SAFE are highly applicable to interests of local stakeholders even with medium to

575 low power computers since SAFE uses Google servers. The tourism sector can also benefit from this snow avalanche deposit inventory, especially the winter sports industry. Furthermore, this method can also be used to prioritize areas for more sophisticated and data-intensive avalanche risk analysis (Keylock et al., 1999). SAFE can be applied by any user throughout mountainous regions of the world as it is designed to be user-friendly, and frequent users can contribute to the robustness of the snow avalanche deposit archive, thus improving recommendations for policy makers.

**Author contributions**

A.C. designed the concept of SAFE method, wrote the Google Engine script and processed the analyses of snow avalanches. A.C. and R.C.S. participated the conception of SAFE and all authors helped interpret the results. D.R.G. contributed to the writing and provided the AKAH dataset of impacted villages by snow avalanches. A.C. and R.C.S. wrote the paper.

**Competing interests**

The authors declare that they have no conflict of interest.

**Acknowledgements**

The authors are very thankful to Nusrat Nahab, Head of Emergency Management Aga Khan Agency for the Aga Khan Agency for Habitat for her support in this study and for sharing information about snow avalanches in Afghanistan. This study was implemented under the ongoing project "Addressing Climate Change in Afghanistan (E3C)" funded by the European Union, in close collaboration with Aga Khan Foundation and Wildlife Conservation Society based in Afghanistan.

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
