# Peer review of "Snow Avalanche Frequency Estimation (SAFE): 32 years of monitoring 1"

_The Cryosphere, 2022_

## Community Comment (CC1)

This paper "Snow Avalanche Frequency Estimation (SAFE): 32 years of remote hazard monitoring in Afghanistan" attempts to produce inventories of avalanche debris using Landsat optical satellite imagery in late spring when snow, bare ground and water are easily distinguishable. The concept of using a long time series of remote sensing data to identify hotspots of avalanche deposition zones and trends in their spatial occurrence is good, but there are many pitfalls with the overall implementation and communication of the work which reduces the impact.

1. The paper requires some major restructuring of the content, starting with the introduction. Throughout the paper I found that information was in the wrong order and/or wrong section. Results were presented already in section 2 (eg. Table 2) and discussions were being made in the results section. This makes the work difficult to follow, even with the flow chart provided. Moreover figures are wrongly labeled (Fig. 10) and have unsatisfactory captions or text to explain what is being shown or how they were produced, color scales are not constant making figures hard to compare (Fig. 6-8).

*Dear reviewer, the authors are very thankful for your comments on our paper. For now we address your comments here, but we are already working on the new version of the manuscript based on your points. Regarding the order of the information, you must be referring to the validation that we indeed presented in the methodology section. We will move it to the results section based on your comment. Regarding Figure 10, we will improve the caption and ensure that the pixels on this map are the pixels were significant temperature trends occurred. As for the colors of figures 6-8, it is unclear exactly what you are referring to here. If you mean that we used different descretations for each of the 4 categories in Fig. 7 (i.e., different binning of total avalanches per village), as well as for number of avalanches per km of road (Fig. 8), then if we normalized each of these it would not clearly show where the major impacts were – a key point of our paper. Thus, we are reluctant to change this as it would obscure the importance of the avalanche impacts.*

2. It seems to me that the authors are basically identifying late season snow patches in valley bottoms close to rivers which they are assuming to be avalanche deposits. This is made quite straightforward by the fact that the regions of interest are snow-free and snow is easily distinguishable by higher NDSI in the Landsat images compared with bare ground or water. This just reduces the problem to a simple thresholding and classification of image pixels into 3 classes, and I fail to see what is state-of-the-art in this approach. Moreover the authors have employed MODIS data to identify the snowline in order to select the dates and regions which are snow-free. MODIS has poorer spatial resolution than Landsat, so why not just use the Landsat data to identify the snowline? I can't see any value in using MODIS vs. Landsat for this purpose.

*Indeed, the NDSI reclassification approach of SAFE is straightforward but the date as well as the region of interest are the key parameters in this model, and to our knowledge, no other studies have adopted this approach before. In the introduction, we reviewed the literature related to avalanche detection using optical, radar, and Lidar data, but none of these studies used the NDSI as we did in SAFE.*

*Regarding the snowline extraction, we used MODIS because of its coverage and its ease of application. Landsat archives can certainly provide snowline maps with higher resolution but the amount of data to be extracted for this purpose is much greater. Moreover, the cloud coverage on Landsat images presents a greater challenge compared to MODIS because there are more tiles to merge from different times, and the coverage is smaller*

*than MODIS. MODIS was used to separate highlands from lowlands across the entire study area and a coarser resolution was acceptable for this purpose.*

3. Throughout the paper the authors emphasise that the approach is based on Landsat data and the use of the google earth engine because it should be used in areas where internet connection is poor. However, they also highlight that the main end-users of such a dataset are stakeholders and decision makers. Are these stakeholders and decision makers likely to be located in remote mountain villages or the main cities (where internet connection is presumably good)? Are local villagers in these mountain environments really likely to be making use of this dataset? I find it hard to believe that knowing where a large avalanche deposit has occurred several months prior to its detection is likely to be of interest to these people.

*Thank you for those comments. It is true that decision makers are potential users of SAFE, but academics and local research institutes can be potential users as well. There is no restriction of use of this model, and this is why SAFE was implemented in Google Engine, to be used in Europe, Central Asia, Andes, in big and well-connected cities, in small cities and villages in remote areas or wherever someone wants to and can use it. That is the point of open-access scripts such as SAFE. Stakeholders and decisions makers in remote mountain villages must not be excluded from this process and, in the mountains of Central Asia and nearby areas, in particular, these stakeholders are very interested and play an active role in hazard monitoring, mainly because of road blockages in winter. The road connections (and therefore the supplies in food, energy, medical supplies, and other items) to these villages are highly affected by snow avalanches and landslides. Hence, there is a critical need for local decision makers to have hazard frequency maps based on long-term data. Moreover, the remoteness of these villages where local stakeholders reside does not mean that they do not have internet access or competent personal to run SAFE, and we know by experience that such models can be easily run in small towns of high Badakhshan in Afghanistan or Tajikistan where this study was conducted. We could have decided to design SAFE as a toolbox (potentially for income) using expensive software, but this would not be useful in the local context where most of the agencies and stakeholders cannot afford expensive software or programs (and Google is a widely used browser that makes Google Engine a user friendly-interface for any stakeholder). Please remember that the high mountains of Central and East Asia are among the poorest regions of the world and residents are very vulnerable to these hazards. This is our primary target area for SAFE, although it certainly can be used in other mountain regions.*

4. As pointed out by reviewer 1 the classification of avalanche size seems quite arbitrary and does not have much meaning when it is being detected late in the season after it has already partly melted out. It would be more meaningful to show for example a histogram of the avalanche size to show what is being detected rather than applying some random size classification to the detected deposits.

*The size classification was actually based on the distribution of our "avalanche surface areas" as shown in Figure 5a. We thought about classifying our deposit zones based on EAWS, 2018 classification, but our surface areas did not match this universal classification. Additionally, it was stated in review 1 that snow avalanches sizes must be classified by volume; however, the two datasets published on EnviDat (https://www.envidat.ch/#/metadata/satellite-avalanche-mapping-validation) were actually classified by area (m²), therefore we retained our size classification based on surface area.*

5. Inconsistent terminology. Avalanche debris/deposits are referred to as "snow packages", "snow patches", "avalanche depositional" in the paper. The authors should use the correct term and use it throughout.

*Thank you for this comment. In the paper we had used the term "depositional zone", but we now made it more consistent in the manuscript.*

6. Poor validation. In section 3 the authors state that over the 32 years of data analysed they identified around 810,000 avalanche deposits using their dataset. However for the calculation of POD and PPV as shown in Table 2 they have ony used 158 deposits observed using Google Earth images. Moreover they do not describe how the validation data were identified (was this done visually or was there some other algorithm used to detect them in these images?). Overall this does not come across as a satisfactory validation dataset with which to evaluate their detections.

*Thank you for your comment. We used 158 snow avalanches because those were visible in some regions during specific years of Google Earth images as explained on line 243: "A total of 158 snow avalanche depositional zones were easily identified in the riparian buffer zones on Google Earth images in 2001, 2003, 2015, 2017, and 2019". And Google Earth images were indeed used to verify the locations of the avalanches predicted by SAFE. The lack of Google Earth imagery over this 32 year period restricted the number of avalanche deposits we could asses.*

*Moreover, in order to delve into the validation, we are currently conducting the comparison between outlined snow avalanches using SPOT-6 images and SAFE results in Switzerland, as recommended by reviewer one.*

**NOTE: We now incorporate Landsat-9 images to SAFE model, which improves the coverage of Landsat archives worldwide. This will be added into the manuscript (Methodology section).**

---

## Author Comment (AC1)

Dear Reviewer,

We are very thankful for your review and comments on our paper about snow avalanche deposition zones in Afghanistan. For now, we have decided to formulate an initial reply to your comments and then will wait for other reviews before submitting a comprehensive response to all review comments along with the revised version of the paper. Nevertheless, we are already working on your suggestions to effectively address your key comments.

1.The state of the art is incomplete. Several publications, very relevant for this topic have to be considered and discussed. In particular the optical mapping with SPOT6 over the Swiss Alps is important (Bühler et al., 2019). But also, several mappings with Sentinel-1 are missing (Leinss et al., 2020; Karas et al., 2021; Vickers et al., 2016). Considering hazard indication mapping new developments allow for applications over very large areas (Maggioni and Gruber, 2003; Barbolini et al., 2011; Bühler et al., 2022) and was even already conducted in Afghanistan (Bühler et al., 2018). Therefore, the introduction has to be overworked including the relevant publications.

Thank you very much for suggesting those relevant references. We will definitely add those references in the Introduction that contribute to the focus of our paper. Regarding Afghanistan, we have not found any published papers that include results of snow avalanches including the paper you noted. Very few, if any, studies have been conducted there on avalanche hazards at a detailed scale.

2. It is essential to clearly communicate what can be expected from the presented approach in terms of accuracy and reliability. First of all, only very large avalanche **debris** can be mapped. Throughout the paper the authors should use this term and not the term avalanche to be clear. An avalanche consists of a release, a transition and a deposition zone. Only the deposition zone can be partially mapped. There are several problems for example if the avalanche debris is covered by soil / rock or wood (The NDSI is reduced and the deposit is not mapped as avalanche). There is now information on how many avalanches deposited onto one mapped deposit. Typically, this happens several times a year. In the river basins there is often complex terrain with a lot of cast shadow leading to missed avalanche debris. All these uncertainties lead to a very limited reliability of the presented approach. Therefore, it is not eligible to draw all the statistics from the mapped debris as the authors do in the results. These statistics are strongly biased and not reliable. Applying them for hazard mapping or the planning of mitigation measures could be very dangerous.

We thank you for your comment related to the terminology used in our paper. Moving forward, we will use the term 'deposition' or 'deposits' to better characterize the focus of our hazard assessment. The term 'debris' is widely used in geomorphology and often denotes the dynamic part of geomorphic processes (e.g., debris avalanches, debris flows); we prefer to use the more appropriate term "deposition" to characterize the 'static' portion of the snow avalanche hazard, which is what we mapped. As for avalanche structure, it was already explicitly articulated that SAFE only maps the deposition zone: "The avalanches are indeed detectable by delineating their depositional zones (not their release or transition zones)" lines 155-156. Although we feel this was already rather clear, we will add other caveats to clarify that were are not dealing with initiation or runout zone – only depositional zone – as this seems to have caused confusion.

As for snow avalanche deposits that may have been missed using NDSI, we agree that SAFE can omit snow avalanche deposits as already acknowledged in line 268: "Another source of error arises when SAFE cannot detect avalanches due to a dark color on the snow surface associated with surface debris or a debris flow on top of the avalanche." However, because of the advantage of our long-term data base, if SAFE misses an event in one year, the model systematically looks at each pixel in every year – in our case, 32 times (i.e., 32 years of data). Thus, frequently impacted

areas will be identified even if events in a few years are missed due to shadows or debris flows. Thus, we disagree that our statistics are 'not reliable'.

Regarding the comment that the application of this model 'could be very dangerous', we completely disagree with this value judgement. SAFE is one model, if not the first attempt, to map areas at avalanche risk across large scales and on over long time periods in that region. As specifically mentioned in the paper, we looked at avalanche deposits on foot slopes where human settlements are most vulnerable to snow avalanches. SAFE does not examine mid- to upper-slope terrain because in mountainous Central Asia, as well as proximate mountain regions, these upper slope areas are not occupied by humans or their activities during winter. Only foot slopes represent an area at risk; high mountain winter recreational activities – e.g., skiing, and other winter tourism activities – are virtually nonexistent in this vast mountain region. What matters in our region, such as Badakhshan, is the location and frequency of avalanche deposits on villages, roads and, to some extent, streams. SAFE is the kind of model that we need here since it can be freely replicated and used with minimal resources to determine which villages and roads are at frequent risk. This is the advantage of our long-term (albeit less accurate than SPOT) database. In our work here on the border of Afghanistan, we frequently experience road blockages and the isolation of villages for several days because of avalanche deposits. This issue is a higher priority in Central Asia and the surrounding mountain regions than detailed mapping of avalanches on upper slopes.

3. To assess the mentioned uncertainties and potential biases we recommend to test the algorithm with the most complete and accurate avalanche dataset mapped with SPOT6 imagery over the swiss Alps in 2018 and 2019 (Bühler et al., 2019; Hafner and Bühler, 2019) https://www.envidat.ch/dataset/spot6-avalanche-outlines-24-january-2018;https://www.envidat.ch/dataset/spot6-avalanche-outlines-16-january-2019. This exercise could bring clarity into very important questions and help to assess the potential of the presented approach.

We would like to thank the reviewer for this recommendation and for sharing these relevant data with us. We will compare the outlined avalanches in 2018-19 with SAFE results on the same area of Switzerland as suggested. Given that the purpose of SAFE is not to map upper slope avalanches, it must be understood that the approaches in these two studies were quite different. However, we can still conduct a comparison between SAFE outputs and the outlined avalanche deposits at the foot slopes only, using the mask we developed for SAFE. Then only, can the two datasets be appropriately compared.

4. The snow avalanche size classification is totally flawed with respect to reality/ methodology. According to the definition of the EAWS (https://www.avalanches.org/standards/avalanche-size/#largeavalanche) size is mostly defined by volume, runout-length and destruction potential: so basically only avalanches larger than size 3 (large to extremely large) have potential to even reach those places where they are later detected with enough snow for it to remain until summer, Additionally, as the authors state they cannot separate single events, a size classification with the same classes as assigned to whole avalanches shortly after their release is nonsensical also as the area covered in gullies usually means a lot more volume than one would think. This makes methodologically no sense as well as everything derived from this (whether as category or as size).

We understand and appreciate your comment about avalanche size classification. Indeed, we unfortunately do not have any data on avalanche volume, and we admit that our terminology 'avalanches size' is confusing. We however believe that the classification of the cumulated avalanches deposits could be relevant information to highlight and rank the most vulnerable areas. Valleys with large 'size' events represent more vulnerable areas impacted by repetitive avalanches than 'small size' events. And if one valley bottom is affected by only one, but a very large

avalanche, SAFE will still be able to identify it as a single large event, since SAFE maps the cumulated avalanche deposits. Therefore, we will change 'avalanches sizes' to 'avalanches surface area' in the text and figures.

It is not clear why only Landsat is used. Sentinel-2 imagery would also be a big help for the presented approach (even though only available from 2015). What about the potential of other systems such as PLANET? This should be discussed.

It would be indeed interesting to run SAFE with other products such as S2 or other products from PLANET. This could be an area to explore in a future paper. For the objective of this study (i.e., a long-term assessment of hazards in valley bottoms related to avalanches), SAFE uses Landsat archives for two clear reasons: (1) those data are open access, which suits the economic context of local research institutes who cannot afford expensive images such as SPOT; and (2) as noted, the objective of this paper was to look at a long term – 32 years – period of avalanches debris and only Landsat archives can achieve this.

---

## Author Comment (AC2)

Dear Reviewer,

The authors are very thankful to your comments. After we submitted a first response to your comments several weeks ago addressing them point by point eliciting significant modifications to the manuscript. All responses to your comments, as well as modifications that we will make to the manuscript are written in red in the text below your comments. We believe that this revised manuscript is an improved version that clarifies and addresses most all of your comments. Based on your review, we redirected the terminology and the concept of our paper from *snow avalanches mapping* to *avalanche deposit zones* mapping. Moreover, the classification of deposits is now based on surface area instead of the confusing term 'size'. The revised version also includes the comparison between SAFE outputs and the dataset you provided in Switzerland; the authors are very thankful for this. We believe that this comparison shows interesting results, where datasets are comparable (deposit zones). This comparison also provided a better view on SAFE applicability in other high mountains regions of the world.

**1. The state of the art is incomplete. Several publications, very relevant for this topic have to be considered and discussed. In particular the optical mapping with SPOT6 over the Swiss Alps is important (Bühler et al., 2019). But also, several mappings with Sentinel-1 are missing (Leinss et al., 2020; Karas et al., 2021; Vickers et al., 2016). Considering hazard indication mapping new developments allow for applications over very large areas (Maggioni and Gruber, 2003; Barbolini et al., 2011; Bühler et al., 2022) and was even already conducted in Afghanistan (Bühler et al., 2018). Therefore, the introduction has to be overworked including the relevant publications.**

Many of these references have now been added to the literature review:

- Lines 61-62 "Vickers et al. (2016) conducted one of the first studies utilizing Sentinel-1 products to detect avalanches debris by developing an unsupervised classification."
- Lines 64-66 "Using TerraSAR-X and Sentinel-1 products, Leinss et al. (2020) mapped avalanches, demonstrating the potential of radar products in snow hazard detection."
- Lines 67-68 "Moreover, a recent study also used SAR products to detect avalanches and demonstrated both the potential and limitations of radar products due to orientation and orbit of the images (Karas et al., 2021)."

In reference to other papers using DEM and GIS technics, we have added the following:

- Lines 67-72: "In addition to optical, radar or Lidar data, other studies used Digital Elevation Models (DEMs) and topographic parameters to determine the influence of terrain on avalanches in Switzerland (Maggioni and Gruber, 2009). Other studies incorporated other parameters such as morphology and vegetation to define potential avalanche zones and ran the Avalanche Flow and Run-out Algorithm to automatically detect potentially affected regions by avalanches (Barbolini et al., 2011). Moreover, the combination of snow measurements (depth) and high resolution DEMs have proved useful in snow hazard detection (Bühler et al., 2018a)."

For the optical state of the art, we have added the following:

- Lines 97-99: "Across a wide area (12,500 km$^2$) individual snow avalanches were manually digitized using high resolution SPOT-6 images (Bühler et al., 2019)."

Moreover, we have added some of these references in Table 4 (line 536) when we compare SAFE accuracy with accuracies of other methods. We thank you for those references that enhance our paper.

**2. It is essential to clearly communicate what can be expected from the presented approach in terms of accuracy and reliability. First of all, only very large avalanche debris can be mapped. Throughout the paper the authors should use this term and not the term avalanche to be clear. An avalanche consists of a release, a transition and a deposition zone. Only the deposition zone**

can be partially mapped. There are several problems for example if the avalanche debris is covered by soil / rock or wood (The NDSI is reduced and the deposit is not mapped as avalanche). There is now information on how many avalanches deposited onto one mapped deposit. Typically, this happens several times a year. In the river basins there is often complex terrain with a lot of cast shadow leading to missed avalanche debris. All these uncertainties lead to a very limited reliability of the presented approach. Therefore, it is not eligible to draw all the statistics from the mapped debris as the authors do in the results. These statistics are strongly biased and not reliable. Applying them for hazard mapping or the planning of mitigation measures could be very dangerous.

We thank you for your comment related to the terminology used in our paper. We have modified snow avalanches to avalanche depositional zones or deposits throughout the manuscript (in red in the text). Moreover, based on your comment, we have decided to change the title as follows: "Snow Avalanche Frequency Estimation (SAFE): 32 years of monitoring remote avalanche depositional zones in Afghanistan". We believe that this title better reflects what SAFE does.

As for avalanche structure, it was already explicitly articulated that SAFE only maps deposition zones: "These zones are indeed detectable by delineating the depositional zones of the avalanches (not their release or transition zones);" lines 175-176.

As for snow avalanche deposits that may have been missed using NDSI, we agree that SAFE can omit snow avalanche deposits as already acknowledged in line 268: "Another source of error arises when SAFE cannot detect avalanches depositional zones due to a dark color on the snow surface associated with surface debris or a debris flow on top of the avalanche." However, because of the advantage of our long-term data base, if SAFE misses an event in one year, the model systematically looks at each pixel in every year – in our case, 32 times (i.e., 32 years of data). Thus, frequently impacted areas will be identified even if events in a few years are missed due to shadows or debris flows. Thus, as we answered in our previous response to this review, we disagree that our statistics are 'not reliable'.

Regarding the comment that the application of this model 'could be very dangerous', as we wrote in the first response to your comment, we completely disagree with this value judgement. SAFE is one model, if not the first attempt, to map areas at avalanche risk across large scales and on over long time periods in this remote, high mountain region. As specifically mentioned in the paper, we looked at avalanche deposits on foot slopes where human settlements are most vulnerable to snow avalanches. SAFE does not examine mid- to upper-slope terrain because in mountainous Central Asia, as well as proximate mountain regions, these upper slope areas are not occupied by humans or their activities during winter. Only foot slopes represent an area at risk; high mountain winter recreational activities – e.g., skiing, and other winter tourism activities – are virtually non-existent in this vast mountain region. What matters in our region, such as Badakhshan, is the location and frequency of avalanche deposits on villages, roads and, to some extent, streams. SAFE represents a very needed model in this region because it can be freely replicated and used with minimal resources to determine which villages and roads are at frequent risk. This is the advantage of our long-term (albeit less accurate than SPOT) database. In our work here on the border of Afghanistan, we frequently experience road blockages and the isolation of villages for several days because of avalanche deposits. This issue is a higher priority in Central Asia and the surrounding mountain regions than detailed mapping of avalanches on upper slopes.

3. To assess the mentioned uncertainties and potential biases we recommend to test the algorithm with the most complete and accurate avalanche dataset mapped with SPOT6 imagery over the swiss Alps in 2018 and 2019 (Bühler et al., 2019; Hafner and Bühler, 2019) https://www.envidat.ch/dataset/spot6-avalanche-outlines-24-january-2018;https://www.envidat.ch/dataset/spot6-avalanche-outlines-16-january-2019. This exercise could bring clarity into very important questions and help to assess the potential of the presented approach.

First of all, the authors would like to thank the reviewer for allowing us access to this very interesting dataset in Switzerland. Hence, we have conducted the comparison between the outputs of SAFE and SPOT-6 outlining method. A totally new paragraph, plus a table, have been added to the manuscript. Based on the comment by Reviewer 2, we have decided to move the validation sections (including SAFE/SPOT comparison) to the beginning of the Results section - sub-sections 3.1 and 3.2. While we believe that the two datasets (SAFE/SPOT) are quite different as explained in the following paragraph, this comparison provides some interesting results. The comparison actually helped us to be more specific about the best use of SAFE. SAFE is more applicable in high mountains (Himalaya, Tien Shen, Hindu Kush, Karakoram, Andes…) where avalanches depositional zones remain longer compared to lower elevation mountains. And, interestingly, these mountain ranges are mostly within impoverished regions. Hence, from the title new version in comment 2, we have modified again the paper title as follows: "Snow Avalanche Frequency Estimation (SAFE): 32 years of monitoring remote avalanche depositional zones in high mountains of Afghanistan". Related to this, we have added two sentences, one in the Introduction and one in the Conclusion, noting that SAFE is more applicable for high mountain regions:

- lines 113-115 "SAFE is applicable in any high mountains of the world, such as Tien Shen, Himalaya, Hindu Kush, Karakoram or Andes, but not restricted to these, where snow avalanches deposits can be detected every year by satellite images for a long time before completely melting."
- and lines 553-555 "Moreover, the application of SAFE in Afghanistan, compared to its application in Switzerland, showed that the script can be applied worldwide, especially in high mountains (above 4000 m) since deposit zones are still detectable in late spring at those elevations."

Moreover, this comparison more precisely elucidates the goal of SAFE, which is mapping avalanche deposits, rather than the continuum from initiation to runout.

The results of the comparison between SAFE/SPOT are presented as follows lines 304-341:

*"3.2 SAFE outputs compared with outlined avalanches using SPOT-6 images*

As a potential method of strengthening our testing of SAFE, outputs of our model were compared with a method that applied a more precise and expensive remote sensing product in Switzerland in 2018 (Bühler et al., 2019). The Swiss area encompassed 12,500 km² where more than 18,000 snow avalanches were manually digitized using very high-resolution products SPOT-6 images (in January 2018). While our dataset is quite different from the Swiss data, the objective of this comparison was to assess how many snow avalanche deposits SAFE could detect compared to the approach using SPOT-6 (Table 3).

**Table 3. Comparison of snow avalanches deposits zones between SAFE outputs (April to June 2018) and manual digitization using high-resolution SPOT-6 images in Switzerland in January 2018\***

| Method | Number of snow polygons | Area of snow polygons (m²) |
|---|---|---|
| SPOT digitization | 7574 | 362,187,741 |
| SAFE detection | 9948 | 494,454,599 |
| Overlapping SPOT-SAFE | 2194 | 223,907,868 |

*SPOT data based on (Bühler et al., 2019)

Importantly, not all avalanches manually digitized on SPOT-6 images were comparable to SAFE results. To make this comparison more consistent, we clipped the outlined avalanches with the valley bottom mask used in SAFE. Following this modification, the SPOT-6 digitization process identified 7574 avalanches deposits in valley bottoms compared with 9948 by SAFE. Overlapping these two datasets, we found that both approaches detected 2194 deposit zones in common. Much of this discrepancy is due

to the timing of SAFE images, which examine deposits that remain in late spring and early summer, whereas SPOT digitization covered only January. The larger number of snow deposits detected by SAFE occur during late season snow avalanches that impact valleys. This suggests that SAFE could not detect all January snow deposits because many of those already melted by the time of SAFE detection (early April to late June in the Swiss case). In addition, optical image quality strongly depends on cloud cover that may cause avalanches to be obstructed. For instance, we could not compare the 2019 SPOT-6 derived dataset in eastern Switzerland (Hafner et al., 2021) due to cloudy images at the end of winter and early spring because these snow avalanches had already melted, implying that SAFE is more suitable for high mountain areas (>4000 m) where snow deposits remain longer in valleys, thus inflicting greater damages and obstructions. Using LANDSAT images, SAFE somewhat circumvents this problem of cloud cover by assessing many years of data (in our case 32 y). However, SAFE does not distinguish individual events and considers overlapping snow deposits as one, in contrast to SPOT-6 which distinguishes these as discrete events. This, in addition to the different methods and spatial resolution difference between SAFE and SPOT, explains the somewhat low number of overlapping snow deposits between SAFE and SPOT. Moreover, the SPOT digitization procedure found a total avalanche area of 362,187,741 m² in January, while SAFE detected 494,454,599 m² of deposits at the end of the avalanche season, including 223,907,868 m² in common. The area detected by SAFE is naturally larger than SPOT-6 since SAFE maps all detectable deposits at the end of the winter. Moreover, SAFE did not detect the small avalanches of January that rapidly melted after they occurred. The polygons extracted by SAFE using Landsat images are obviously coarser than those outlined with SPOT-6 images, which partly explains the low number of overlapping snow deposit zones, but a much more comparable detected area (62%) between the two methods. Much of the discrepancy is related to SAFE's inability to detect individual events and missing deposits that rapidly melt (mostly from the early winter snow avalanches), as well as the very different resolution of these products."

**4. The snow avalanche size classification is totally flawed with respect to reality/ methodology. According to the definition of the EAWS (https://www.avalanches.org/standards/avalanche-size/#largeavalanche) size is mostly defined by volume, runout-length and destruction potential: so basically only avalanches larger than size 3 (large to extremely large) have potential to even reach those places where they are later detected with enough snow for it to remain until summer, Additionally, as the authors state they cannot separate single events, a size classification with the same classes as assigned to whole avalanches shortly after their release is nonsensical also as the area covered in gullies usually means a lot more volume than one would think. This makes methodologically no sense as well as everything derived from this (whether as category or as size).**

We understand and appreciate your comment about avalanche size classification. Indeed, we unfortunately do not have any data on avalanche volume, and we admit that our terminology 'avalanches size' is confusing. We however believe that the classification of the cumulative avalanche deposits could be relevant information to highlight and rank the most vulnerable areas. Valleys with large 'size' events represent more vulnerable areas impacted by repetitive avalanches than 'small size' events. And if one valley bottom is affected by only one, but a very large avalanche, SAFE will still be able to identify it as a single large event, since SAFE maps the cumulated avalanche deposits. In addition, we noticed that the avalanches mapped using SPOT-6 images in Switzerland (https://www.envidat.ch/dataset/spot6-avalanche-outlines-24-january-2018;https://www.envidat.ch/dataset/spot6-avalanche-outlines-16-january-2019) were actually also classified by size (m²), so the point about the 'flawed classification' raised by the reviewer is confusing and somewhat contradictory compared to this Swiss data. To address this issue, we have changed 'size' to 'surface area' in the manuscript wherever needed (text and Figure 11).

**5. It is not clear why only Landsat is used. Sentinel-2 imagery would also be a big help for the presented approach (even though only available from 2015). What about the potential of other systems such as PLANET? This should be discussed.**

As noted in our initial response to this comment, we agree that it would indeed be interesting to run SAFE with other products such as S2 or other products from PLANET. This could be an area to explore in a future paper. However, to address the objective of this study (i.e., a long-term assessment of hazards in valley bottoms related to avalanches), SAFE uses Landsat archives for two obvious reasons: (1) these data are open access, which suits the economic context of local research institutes and practitioner in this greater region who cannot afford expensive images such as SPOT; and (2) as noted, the objective of this paper was to look at a long-term – 32 years – period of avalanches deposits to assess frequencies of potential damages, and only Landsat archives can achieve this.

---

## Author Comment (AC3)

Dear reviewer,

We wish to thank you for your constructive and helpful comments on our manuscript about SAFE in Afghanistan. A few weeks ago, we had provided a first response to your comments. Now we address all your comments in the context of the modifications we plan to incorporate in the revised version of the manuscript (in green). Based on your comments, we have changed the terminology of the paper from *snow avalanches* to *avalanches deposit zones*. In addition, the figures were improved, as requested. We also believe that the paper shows a better understanding of the applicability of SAFE. Indeed, we have emphasized the interest of locals in models such as SAFE and its applicability in remote high mountains. Moreover, as pointed by both reviews, we have significantly improved the validation section. Please, find our detailed response below.

This paper "Snow Avalanche Frequency Estimation (SAFE): 32 years of remote hazard monitoring in Afghanistan" attempts to produce inventories of avalanche debris using Landsat optical satellite imagery in late spring when snow, bare ground and water are easily distinguishable. The concept of using a long time series of remote sensing data to identify hotspots of avalanche deposition zones and trends in their spatial occurrence is good, but there are many pitfalls with the overall implementation and communication of the work which reduces the impact.

1. The paper requires some major restructuring of the content, starting with the introduction. Throughout the paper I found that information was in the wrong order and/or wrong section. Results were presented already in section 2 (eg. Table 2) and discussions were being made in the results section. This makes the work difficult to follow, even with the flow chart provided. Moreover figures are wrongly labeled (Fig. 10) and have unsatisfactory captions or text to explain what is being shown or how they were produced, color scales are not constant making figures hard to compare (Fig. 6-8).

The authors are very thankful for your comments on our paper. As specified in our first response few weeks ago, we have restructured the paper accordingly. Now the validation section is at the beginning of the Results, in Sections 3.1 and 3.2.

Regarding Figure 10, we have improved the caption as follows lines 447-450: "Figure 10. a, Map of areas with significant increases in monthly land surface temperatures in the Amu Panj Basin based on MOD11C3 products from 2000 to 2021; b, Geographical shift of avalanche depositional zones: mean longitude and latitude of avalanche deposits each year since 1990 show evidence of a movement to the northeast due to increasing winter temperature in mountainous areas.". To explain how the map was produced, we also added the following lines 437-438: "However, the slope was calculated and a Mann-Kendal test was applied for each pixel of the land surface temperature images (MOD11C3)."

Figures 6-8 were redesigned based on your comment and the label colors are now consistent amongst all maps (sub-catchments, villages and roads).

2. It seems to me that the authors are basically identifying late season snow patches in valley bottoms close to rivers which they are assuming to be avalanche deposits. This is made quite straightforward by the fact that the regions of interest are snow-free and snow is easily distinguishable by higher NDSI in the Landsat images compared with bare ground or water. This just reduces the problem to a simple thresholding and classification of image pixels into 3 classes, and I fail to see what is state-of-the-art in this approach. Moreover the authors have employed MODIS data to identify the snowline in order to select the dates and regions which are snow-free. MODIS has poorer spatial resolution than Landsat, so why not just use the Landsat data to identify the snowline? I can't see any value in using MODIS vs. Landsat for this purpose.

As noted in our initial response, indeed, the NDSI reclassification approach of SAFE is straightforward, but the date as well as the region of interest are the key parameters in this model. To our knowledge, no previous studies have adopted this approach. In the introduction, we reviewed the literature related to

avalanche detection using optical, radar, and Lidar data, but none of these studies used the NDSI as we did in SAFE.

Regarding the snowline extraction, we used MODIS because of its coverage and its ease of application. Landsat archives can certainly provide snowline maps with higher resolution but the amount of data to be extracted for this purpose is much greater. Moreover, the cloud coverage on Landsat images presents a greater challenge compared to MODIS because there are more tiles to merge from different times, and the coverage is smaller than MODIS. MODIS was used to separate highlands from lowlands across the entire study area and a coarser resolution was acceptable for this purpose.

3. Throughout the paper the authors emphasise that the approach is based on Landsat data and the use of the google earth engine because it should be used in areas where internet connection is poor. However, they also highlight that the main end-users of such a dataset are stakeholders and decision makers. Are these stakeholders and decision makers likely to be located in remote mountain villages or the main cities (where internet connection is presumably good)? Are local villagers in these mountain environments really likely to be making use of this dataset? I find it hard to believe that knowing where a large avalanche deposit has occurred several months prior to its detection is likely to be of interest to these people.

Thank you for those comments. In our first response, we noted that there is indeed local interest and concurrent local knowledge related to running SAFE (see RC2). To emphasize this point, we have added the following sentences to the conclusion lines 561-565: "Moreover, villages of high mountains such as in Afghanistan are strongly highly dependent on roads connections to provide necessary food, energy, medical supplies, and life-support items, especially in winter. It is therefore critical for local decision makers to assess the frequency of road blockage by avalanche deposits. Thus, open-access and user-friendly tools such as SAFE are highly applicable to interests of local stakeholders even with medium to low power computers since SAFE uses Google servers."

4. As pointed out by reviewer 1 the classification of avalanche size seems quite arbitrary and does not have much meaning when it is being detected late in the season after it has already partly melted out. It would be more meaningful to show for example a histogram of the avalanche size to show what is being detected rather than applying some random size classification to the detected deposits.

It was indeed stated in Review 1 that snow avalanche sizes must be classified by volume; however, the two datasets published on EnviDat (https://www.envidat.ch/#/metadata/satellite-avalanche-mapping-validation) were actually classified by area (m²), therefore we retained our size classification based on surface area. However, as noted in our response to Reviewer 1, we changed 'size' to 'surface area' wherever needed, for more clarity.

5. Inconsistent terminology. Avalanche debris/deposits are referred to as "snow packages", "snow patches", "avalanche depositional" in the paper. The authors should use the correct term and use it throughout.

Thank you for this comment. In the paper we had used the term "depositional zone", but we now made it more consistent in the revised manuscript.

6. Poor validation. In section 3 the authors state that over the 32 years of data analysed they identified around 810,000 avalanche deposits using their dataset. However for the calculation of POD and PPV as shown in Table 2 they have ony used 158 deposits observed using Google Earth images. Moreover they do not describe how the validation data were identified (was this done visually or was there some other algorithm used to detect them in these images?). Overall this does not come across as a satisfactory validation dataset with which to evaluate their detections.

Thank you for your comment. We used 158 snow avalanches because those were visible in some regions during specific years of Google Earth images as explained on line 243: "A total of 158 snow avalanche depositional zones were easily identified in the riparian buffer zones on Google Earth images in 2001, 2003, 2015, 2017, and 2019". And Google Earth images were indeed used to verify the locations of the avalanches predicted by SAFE. The lack of Google Earth imagery over this 32-year period restricted the number of avalanche deposits we could assess. For more clarity on this matter, we have added the following lines 268-272: "No other Google Earth images were available during the last 32 years in Afghanistan, therefore the comparison between SAFE and the true events was conducted with those available 158 deposit zones. These 158 deposits were extracted from Google Earth and stacked with SAFE outputs. SAFE deposits were considered as valid when the two datasets were overlapped at the same location.".

Moreover, to improve the validation, we have completed the comparison between outlined snow avalanches using SPOT-6 images and SAFE results in Switzerland, as recommended by Reviewer 1. Results are now in section "3.2 SAFE outputs comparison with outlined avalanches using SPOT-6 images".

---

## Author Response (AR2)

Dear reviewer,

First of all, we would like to thank you for your comments and for considering our paper for publication. You will find in the following paragraphs, our responses to your comments. They are in blue in the manuscript.

**Thanks for taking up my input and comparing your results to our 2018 data from Switzerland. I am aware that it is not straight forward to compare these two approaches. However, I think this is very important to get an idea about the strength and weaknesses of your algorithm. I think it is important to have a Figure, illustrating the comparison between the two products. The dataset you used can also be cited itself and has a DOI (Hafner, E. and Bühler, Y.: SPOT6 Avalanche outlines 24 January 2018, EnviDat [dataset], doi:10.16904/envidat.77, 2019.)**

Once again, we would like to thank you for sharing those data with us. We agree that one illustration of the comparison is needed and have decided to add a new figure (F4) line 321. Moreover, we have added the following lines to comment on this figure (lines 317-320):

"Figure 4 shows an illustration of this comparison. It appears that the deposit zones detected by SAFE are in line with SPOT6 outlined avalanches. The later however covers the entire avalanches while SAFE only detects, automatically, the deposit zones.

[Figure]

**Figure 4. An illustration of the comparison between automatic detection of deposit zones using Landsat archives in SAFE and manually outlined snow avalanches (from origin to deposit zones) using SPOT6 images in Switzerland"**

As for the reference dataset, we have added your suggestion line 312-314, where we cited Hafner et al., 2021: "As a potential method of strengthening our testing of SAFE, outputs of our model were compared with a method that applied a more precise and expensive remote sensing product in Switzerland in 2018 (Bühler et al., 2019b; Hafner and Bühler, 2018)."

**P1L10: The statement "only local maps to estimate snow avalanche risk have been produced" is not true! Another example would be in this publication: Soteres, R. L., Pedraza, J., and Carrasco, R. M.: Snow avalanche susceptibility of the Circo de Gredos (Iberian Central System, Spain), Journal of Maps, 16, 155-165, 10.1080/17445647.2020.1717655, 2020. Just now our most recent paper published producing such maps for the entire canton of Grisons in Switzerland: Bühler, Y., Bebi, P., Christen, M., Margreth, S., Stoffel, L., Stoffel, A., Marty, C., Schmucki, G., Caviezel, A., Kühne, R., Wohlwend, S., and Bartelt, P.: Automated avalanche hazard indication mapping on a statewide scale, Nat. Hazards Earth Syst. Sci., 22, 1825-1843, 10.5194/nhess-22-1825-2022, 2022 (https://nhess.copernicus.org/articles/22/1825/2022/nhess-22-1825-2022.html). Please add this publication where you cite Bühler et al. 2018a as this publication is the further development of this approach.**

Thank you for sharing those relevant references. First of all, we have added Soteres et al., 2020 to the manuscript where we mention the DEM maps lines 101-103: "Terrain parameters like slope gradient and curvature have also been added to the avalanche detection process using DEMs (Soteres et al., 2020) combined with Landsat-8 images (Bühler et al., 2018b; Singh et al., 2019)."

In order to avoid any confusion and based on your comments, we have removed "only local maps to estimate snow avalanche risk have been produced" from the abstract.

As for the paper in Grisons, we have added the reference as suggested lines 71-72: "Moreover, the combination of snow measurements (depth) and high resolution DEMs have proved useful in snow hazard detection (Bühler et al., 2022, 2018a)."

**p11L299: you say SAFE can be considered as conservative and robust. This might be true but this gets a problem if you calculate frequency (e.g. Fig. 5) from it as it misses a large part of the avalanches in high winter (but these avalanches are at least as dangerous as the avalanches in spring). This point has to clarified and the limitations have to be described very clear in particular in connection with avalanche frequency.**

Thank you for this comment. We had already written the following sentence in the previous version of the manuscript, clearly stating that SAFE cannot detect early winter deposits line 353-354: "Much of the discrepancy is related to SAFE's inability to detect individual events and missing deposits that rapidly melt (mostly from the early winter snow avalanches)".

However, in order to further clarify, we have added the following sentence as suggested in lines 302-307: "Moreover, it should be understood by the users that another limitation is that SAFE does not detect early winter avalanche deposits due to melting and snow coverage on and around the snow deposit, which might affect the deposits frequency estimations. However, based on our findings, SAFE can be considered as a conservative, yet robust and efficient tool to automatically identify snow avalanche depositional zones in very remote areas and can be applied in any mountainous region."

**P10L271: how much do they need to overlap to count as OK? Only by one pixel? This should be explained maybe also with a figure.**

We considered the snow deposits extracted from SAFE as reliable when more than 50% of the polygons was overlapping the actual deposits on Google Earth Image. In order to improve the understanding; we have added the following lines to the manuscript (lines 274-276): "Deposits identified by SAFE were considered valid when the two datasets overlapped at the same location and when more than half of the polygon surfaces extracted from SAFE overlapped the actual deposits visible on Google Earth images."

We haven't added a new figure since we believe that there are already many figures in our paper, moreover, we believe that the new figure 4 about the comparison between SAFE and Swiss data already shows how SAFE outputs look compared to actual avalanches manually outlined.

**P25L542: I would add "in spring and early summer" only then the avalanches are mapped.**

We have modified the text accordingly line 557: "SAFE can be considered as a universal approach to assess snow avalanche depositional zones in spring and early summer where ground data are very limited, such as in the Afghan mountains."